# Perceived Training Needs of the Informal Caregivers of Older Adults: A Cross-Sectional Study

**DOI:** 10.3390/healthcare12232369

**Published:** 2024-11-26

**Authors:** Qianqian Chen, Huimin Zhang, Suwei Yuan, Wenwei Liu, Tongzhou Lyu

**Affiliations:** 1School of International and Public Affairs, Shanghai Jiao Tong University, Shanghai 200030, China; chenqianqian123@sjtu.edu.cn; 2School of Government, Central University of Finance and Economics, Beijing 100081, China; zhanghuimin25@163.com; 3Ren Ji Hospital, Shanghai Jiao Tong University School of Medicine, Shanghai 200127, China; yuansuwei@renji.com; 4College of Philosophy, Law and Political Science, Shanghai Normal University, Shanghai 200234, China; 5School of Politics and International Relations, East China Normal University, Shanghai 200241, China

**Keywords:** informal caregiver, training needs, health oriented, personalization

## Abstract

Background: Due to the enormous caregiving burden faced by informal caregivers, providing appropriate skills training has become an important supporting strategy in many countries/regions. Understanding caregivers’ training needs is instrumental in designing effective training intervention programs, which are expected to reduce the caregiving burden of informal caregivers and avoid the health deterioration associated with caregiving. This paper aims to explore the potential training needs of informal caregivers in Shanghai, and to identify the factors associated with these perceived training needs. Methods: A total of 196 eligible informal caregivers participated in this survey. A multivariate analysis was conducted to explore the factors associated with informal caregivers’ perceived training needs. Results: 86.7% (*N* = 170) of the caregivers reported at least one need for targeted training activity, and 62.7% (*N* = 123) of them identified two or more training needs. The top three activities requiring training included the following: self-care skills; safety supervision; and functional rehabilitation. The factors associated with various training needs included the health status of the care recipient, complementary caregiving support, caregiving stress, and the personal attributes of the informal caregiver. The technical skills training needs were more related to the care recipients’ health status (e.g., dependency level, disease progression) and formal care support resources. Conversely, the intangible skills training needs were more sensitive to caregiver attributes (e.g., gender, age, and education level). Conclusions: A personalized training strategy and early-stage intervention program are critical to providing effective support to informal caregivers. The potential implications are to raise awareness of the importance of skills training for informal caregivers, and to inform the implementation of effective training strategies for improving the quality of informal care and the well-being of informal caregivers in China.

## 1. Introduction

Population aging, greater life expectancy, and chronic illnesses have increased long-term care needs [1], and the current formal care system can hardly cope with these needs due to the lack of trained professionals. In this context, informal care has become the primary means of caregiving and has begun to play a pivotal role in the current long-term care system. However, unlike professional formal care providers, informal caregivers typically enter their caregiving role based on specific cultural values, societal norms, and perceived expectations, rather than career choices [2]. They have to address a far-reaching domain of complex and sensitive health care issues, but with little background or training in providing quality care to their loved ones [3,4]. When these high-intensity caregiving tasks overlay with significant caregiving skill gaps, caregivers can experience substantial stress and markedly decrease their physical and mental health [5,6,7]. Actually, numerous studies have demonstrated the correlation between informal caregiving and health status [8]. Among them, the Stress Process Model proposed by Pearlin et al. is widely used [9]. Specifically, the complex caregiving needs of care recipients are often considered as a primarily stressor, leading to a subjective caregiving burden for caregivers, either directly or through influencing their caregiving behaviors. When informal caregivers are burdened or overloaded for a long period of time, it can undoubtedly have a significant negative impact on their physical and mental health. However, all the steps in this process are influenced by pre-existing personal characteristics (e.g., caregiving experience, skills) and concurrent buffering factors (e.g., social support) [9,10,11]. Accordingly, it is imperative to provide effective support interventions to enable informal caregivers to adequately cope with their caregiving challenges. The provision of caregiving skills training is widely recognized as one of the most effective support interventions [12].

Many developed countries/regions have incorporated skills training into caregiver support programs. For instance, the United States introduced its first federally funded program, called “The National Family Caregiver Support Program (NFCSP)” in 2000, which implemented a series of targeted skills training items to help caregivers better provide quality care [13]. Similar nationwide initiatives also include the “Commonwealth Home Support Program (CHSP)” in Australia, the “Caring in the Home” program in Iceland, and so on. Extensive academic literature has also shown that quality skills training holds great promise in improving the situation of informal caregivers, i.e., those who have received certain training tend to exhibit less physical and mental health impairment, and higher self-perceived value [14]. In addition, such training is expected to improve the health status of care recipients [15] and increase their ability to remain safely in their homes and communities [16]. Finally, it is one of the effective ways to provide sustainable informal care, along with reducing the pressure and potential costs of formal care [17].

Given the value of well-trained informal caregiving, exploring and analyzing informal caregivers’ training needs so as to better understanding their preferences and characteristics is particularly important. For the identification of training needs, most early research has resorted to using a professional assessment scale and mainly focus on the perspectives of third-party assessors or medical professionals [18,19]. However, due to a lack of attention to informal caregivers’ perspectives, this kind of traditional evidence-based assessment may not adequately reflect their true training needs, which may exacerbate bias in their policy support. In this regard, more research has underlined the importance of subjective perceived training needs [20]. Related studies have found that the perceived training needs were broad and extensive, ranging from medical/nursing skills, domestic skills, coordination, information matching, communication, the use of smart devices, and so on [21,22,23]. But the prominence of each training needs, when reported, differed across studies. For example, Reinhard and Feinberg (2015) found that caregivers were increasingly expecting medically oriented training items, such as skills in peripherally inserted central catheters, feeding tubes, and surgical drains, as well as ostomy or wound care [24]. Some studies have also concerned the importance of the influencing factor analysis of training needs as a pre-requisite to the matched delivery of tailored caregiving skills training [25,26]. A consistent finding was that the training needs of informal caregivers are not only related to the care recipient’s dependency level and disease progression, but are also largely associated with the characteristics of the informal caregiver and caregiving, such as the caring time, the relationship with the care recipient, formal education, the caregiving experience, and the perceived health status [22,27]. Unfortunately, there is a paucity of empirical evidence regarding the relation between training needs and the availability of complementary caregiving supports. Longacre (2008) found that the propensity of receiving training was higher among caregivers who used other care enabling strategies (i.e., information resources, support group, paid or outside resources) [26]. Nevertheless, this study did not delve into the relationship between supplemental support resources and varied training need categories.

In China, informal care comprises a significant part of long-term care; however, there is a lack of high-quality scientific evidence on the perceived training needs analysis. Studies primarily examining caregivers’ needs are missing as needs are mostly extracted in studies covering other issues, such as care burden. Even at the practical level, supportive policies directly targeting informal caregivers remain limited. Instead, nationwide initiatives prefer to support them indirectly by providing financial and service assistance to the care recipients [28]. Examples include the provision of living allowances, home care services, and long-term care insurance for older adults.

As the ‘oldest’ city in China, Shanghai has taken the lead in providing training support for informal caregivers with a program, called “The Older Adult Helping Program”, in 2018. By 2021, the training has exceeded 25,880 participants. Unfortunately, the social organizations undertake this program according to the training checklist provided by the municipal government. There was a lack of systematic training needs assessment in the design and implementation of this program, especially a survey of caregivers’ subjective training needs. As a result, this could lead to inaccurate assessments of informal caregiver support needs and incorrect assumptions about their willingness and capacity to provide levels of care, and finally, the program would deviate from its purpose [22].

This study, to the best of the authors’ knowledge, provides the first empirical research evidence of informal care training needs in Shanghai, China. The objectives of our research are two-fold: (1) to identify the most required care training tasks for informal caregivers in Shanghai; and (2) to explore the factors associated with different training needs, with a particular focus on the association between perceived training needs and various care stressors, as well as complementary caregiving supports (e.g., formal care and other informal care).

Our findings are expected to provide an important reference value for policy makers to design effective caregiving training strategies as well as rationalize resource allocation. Furthermore, appropriate training interventions hold great promise in improving the health status of both informal caregivers and care recipients, increasing the sustainability of informal care, and alleviating the burden on health care systems.

## 2. Materials and Methods

### 2.1. Participants and Sample Size

Since the first iteration of the “The Older Adult Helping Program” was implemented in Shanghai in 2018, four iterations of the program have been consecutively conducted between then and 2021. In this study, a multi-stage cluster random sampling method was used to randomly select 10 sub-districts out of the 45 sub-districts where the program was implemented in 2021 as the investigation sites.

All the informal caregivers of the older adults who engaged in the training program during 2021 were invited to participant in the survey. Considering the differences in the populations cared for by the informal caregivers and the cognitive ability to answer questions related to caregiving training needs, several requirements were established for the inclusion of the respondents. The informal caregiver should be the following: (a) 16 years old or older; (b) the primary caregiver, who has spent at least 1 h per day performing care tasks, or who has provided care consistently for at least 1 month (a minimum of 1 time per week during that month); (c) caring for older adults with any ADL/IADL impairment; (d) not registered, or financially compensated by any formal for-profit institutions.

After filtering out 27 informal caregivers based on the inclusion criteria, a total of 268 eligible informal caregivers voluntarily participated in the survey and, among them, 196 (73.13%) valid samples were obtained for this research. Specifically, 72 respondents were excluded for non-responsiveness of the training needs question (*N* = 39), no report of specific functional impairment of the care recipients (*N* = 10), and other incomplete information (*N* = 17). The respondents who reported caregiving over 16 h/d (*N* = 6) were also excluded to avoid over-pledging bias (then informal caregivers would have less than 8 h/d for rest).

Sample size calculation was conducted with reference to the Kendall sample estimation method to ensure enough data were collected, and the results for the reasonable sample size needed (112 ≤ *N* ≤ 250) indicated that our sample was adequate.

### 2.2. Data Collection

A questionnaire was developed and designed to obtain data, with information including sections on the sociodemographic information about the informal caregivers and care recipients, caregiving-related information, and perceived training needs content.

The survey was based on a government-sponsored evaluation of the “The Older Adult Helping Program”, and informal caregivers were invited to be interviewed at their homes or training centers with the assistance of the training organizations. Following the timeframe of the program evaluation, all the surveys were performed between 3 December 2021 and 23 January 2022. Meanwhile, face-to-face questionnaires were implemented to fully understand the potential training needs of the informal caregivers and to improve the validity and reliability of the survey [22].

Prior to the survey, the research assistant explained the details and purpose of the study to all the informal caregivers, and informed consent was obtained from all the eligible respondents. To avoid over-pledging bias, the surveys were administered without the presence of a third party. On average, it took the respondents 20–30 min to complete the investigation.

### 2.3. Variables and Measures

#### 2.3.1. Identified Training Needs

Identifying possible training content is a prerequisite for effectively assessing training needs. We drew on the training content framework of the Outcome and Assessment Information Set (OASIS). OASIS is a standardized care recipient assessment completed by home health clinicians [29], which has gradually been employed in evaluating the unmet training needs of informal caregivers [25,27]. The identified training contents comprised medical procedures, equipment management, medication management, self-care tasks, safety supervision, household chores, and communication. A 6-member consulting team (including three trainers from informal care training institutions and three experts in the field of long-term care) first translated, localized, and revised the training content based on their professional knowledge and practical experience. Additionally, a training framework with seven categories was initially developed, which included medical/nursing skills, self-care skills, functional rehabilitation, safety supervision, household chores, social/emotional skills, and stress management. Thereafter, a further pilot survey based on 50 informal caregivers was conducted to verify the reliability of the content, and an open box was provided where the participants were asked to report any other self-identified training needs. Interestingly, in the pilot survey, no participant reported “household chores” skills as a training need; instead, “smart devices utilization” skills training was repeatedly mentioned in the open box (nearly 16%, *N* = 8). Therefore, in the second round of consultation, all the experts reached a consensus to replace “household chores” skills with “smart devices utilization” skills in the training content.

A final set of seven training activities were identified, including the following four technical skills and three intangible skills, respectively:Medical/nursing skills (medical procedures, such as wound dressings; equipment management, such as intravenous/infusion equipment; and managing and taking oral, inhaled, or injectable medications);Self-care skills (getting in and out of beds and chairs, dressing, bathing, eating/feeding, toileting, dealing with incontinence/diapers);Functional rehabilitation (plyometrics, joints, cognitive training);Safety supervision (monitoring and management of crisis situations to ensure elder’s safety, such as getting lost, falling, choking, getting burned);Social and emotional skills (communicating with the elderly, adjusting to the elder’s emotions, acting as a communication intermediary between the elderly and medical workers);Stress management (relaxation exercises, stressful event responses, time management, care plan development);Smart device utilization (smart monitoring devices, rehabilitation devices, online tools for emergency assistance, car booking, food ordering, medical appointments).

The perceived training needs of the informal caregivers were obtained via direct questioning. The question posed was as follows:


*Which of the following training activities would you consider necessary to be trained so as to help better manage daily caregiving tasks?*


A training need was considered to exist when any of the subcategory items of a training activity were necessary to provide the training. Each training activity was a dichotomous variable, with 1 indicating “yes” and 0 indicating “no”.

#### 2.3.2. Caregiving-Related Information

Considering the potential association between various training needs and caregiving stressors, two types of caregiving stressors were further included, i.e., physiological stressor and emotional stressor. For the physiological stressor, caregiving time was adopted as a proxy variable, referring to the experience of the existing studies [11]. The respondents were asked about the average days of care per week and the hours of care per day in the past month. The weekly caregiving time was obtained by multiplication, and a z-score transformation was further applied. For the emotional stressor, the relationship with the care recipient was selected as the proxy variable. It has been demonstrated that caregivers’ identity constructs and relational connections as an emotional stressor can drive them to be more likely to experience greater psychological burden [5]. The relationship with the care recipient included spouse, children (in law), and other (another family member or friend/neighbor/acquaintance/volunteer). It was operationalized to dummy variables, with “spouse” as the reference category.

Moreover, the availability of complementary caregiving support may likewise be associated with training needs. We considered two sources of support. The first source was the use of formal care support, i.e., professional caregiver support from the government or the market. In particular, in Shanghai, the home care services financially guaranteed by the government consisted of two main types: home care in the form of service vouchers implemented since 2004, and home care covered by the long-term care insurance implemented since 2016. Informal caregivers were asked whether they had (1) or had not (0) received any of these formal care supports during their caregiving period. Other informal care support was the second source of support. This measure indicated whether the care recipient had support from nonprofessional caregivers other than the one interviewed, including family members, friends, neighbors, acquaintances, and volunteers. Similarly, a value of 1 was assigned when the respondent had received any of the other informal support, and a value of 0 otherwise.

#### 2.3.3. Individual-Level Characteristics

Socio-demographic and socio-economic data related to the informal caregivers and care recipients were obtained, including gender, age, squared age, marital status (married/cohabiting, widowed, all others), education, and income (monthly). The education was categorized by the years of schooling as “<7 years”, “7–12 years”, and “>12 years”, which represented the primary, secondary, and higher education level in China’s education system, respectively. The income was classified according to the average pension in Shanghai in 2021 (approximately 4000 CNY/monthly), comprising “<4000”, “4000–6000”, and “>6000”. Moreover, we also included the informal caregivers’ work status (employed vs. unemployed/retired) and living arrangement (co-residence vs. live apart). Given the possible non-linear relationship between the category variables and the dependent variables, the gender, marital status, education, income, work status, and living arrangement were converted into n-1 dummy variables in the correlation analysis.

Considering the significant correlation between the care recipient’s health status and the training needs of the informal caregivers, several evaluation indicators were further incorporated. The Katz Index [30] of the activities of daily living (ADL) and the Lawton Scale [31] of the instrumental activities of daily living (IADL) were collected to measure the disability level of the care recipient, respectively. The ADL scale consisted of 6 items centered primarily on basic physical needs: eating, dressing, continence, toileting, bathing, and indoor ambulating. The IADL scale consisted of 8 more complex activities related to independent living in the community, including the ability to use the telephone, shopping, food preparation, housekeeping, laundry, their mode of transportation, responsibility for one’s own medications, and their ability to handle finances. The informal caregivers were asked whether the care recipient they cared for did (1) or did not (0) have any difficulty with these activities. All the item scores were summed separately to obtain the ADL (scale from 0 to 6; Cronbach’s alpha of 0.744) and IADL scores (scale from 0 to 8; Cronbach’s alpha 0.753). As continuous variables, a higher score means a higher level of disability. The Charlson Comorbidity Index (CCI) was employed as an indicator of the medical condition of the care recipient [32]. The CCI of a specific disease ranges from 1 to 6. It quantifies comorbidities according to the severity of the illness and the intensity of the required resources, in which a higher CCI indicates a more serious medical condition. In addition, since care for different disorders may vary markedly and result in differences in caregiving training needs [20], dementia was included as a separate indicator.

#### 2.3.4. Statistical Analysis

All the statistical analyses were performed using IBM SPSS Statistics version 25.0 (IBM Corp., Armonk, NY, USA). Descriptive statistics analysis was used for describing the potential predictor variables. Specifically, the descriptive variables were expressed as means and standard deviations (SD), and the categorical variables were reported as numbers and percentages. A univariate analysis was carried out to identify the factors associated with each of the care training needs. Depending on the type of variables in the correlation analysis, the chi-square test and the Mann–Whitney U-test were used, respectively. To avoid including confounding factors or missing important factors, a moderate parameter selection of a *p* value < 0.200 was adopted in the univariate analyses (see Table A1) [33], which is a broad selection of existing studies. The variables significantly associated with the training needs in the univariate analysis were included in the binary logistic regression model. Regression was performed via a stepwise approach, and the independent variables were screened by the likelihood ratio test with a maximum partial likelihood estimation to obtain the final regression model. Nonsignificant predictor variables were sequentially eliminated, and the models were re-estimated until all the variables remaining in the model showed significant association with the outcome. A confidence level of 95% was adopted in the multivariate analysis.

## 3. Results

### 3.1. Characteristics of the Participants

Among the 196-participant valid sample, female informal caregivers accounted for 66.3% (*N* = 130), which was nearly twice as many as the male caregivers (Table 1). The average age of the caregivers was 61.18 years old (*SD* = 13.16), most of which were married (*N* = 161, 82.1%) and had received 7–12 years of education (*N* = 128, 65.30%). Employed, retired, and unemployed caregivers comprised 25.5% (*N* = 50), 67.9% (*N* = 133), and 6.6% (*N* = 13), respectively. In terms of income, half of the caregivers had a monthly income level of 4000 CNY or less (*N* = 98, 50.0%), with 33.7% (*N* = 66) and 16.3% (*N* = 32) earning 4000–6000 CNY and above 6000 CNY, respectively. Notably, 67.9% (*N* = 133) of the respondents reported living with the care recipients, of which child relationships accounted for 49.6% (*N* = 66). This implies that inter-generational cohabitation is still the dominant residence pattern for most households in China. In addition, 52% of the caregivers surveyed were children of the care recipients (*N* = 102), while spouses were slightly less represented than children at 30.1% (*N* = 59). On average, informal caregivers spent approximately 28.02 h (*SD* = 19.215) per week caring for the elderly. Nearly 89.8% (*N* = 176) reported receiving any type of supplementary care support, of which 87.2% (*N* = 171) received other informal care support, and 52% (*N* = 102) received formal care support.

In terms of the care recipients, most were female (*N* = 122, 62.2%) and either married (*N* = 92, 46.9%) or widowed (*N* = 87, 44.4%). The mean age of the care recipients was 78.76 years (*SD* = 9.847), and most of them had a monthly income below 4000 CNY (*N* = 124, 63.3%). The percentage of care recipients’ education level below 7 years, 7–12 years, and 12 years above was 37.8% (*N* = 74), 52.6% (*N* = 103), and 9.7% (*N* = 19), respectively. Regarding disability level, the mean score of the ADL and IADL was 2.02 (*SD* = 1.614) and 4.37 (*SD* = 1.709), respectively. The mean value of the CCI was 1.72 (*SD* = 10.981), and 56.1% (*N* = 120) of the elderly had a CCI above 2, indicating that comorbidity was a common feature of the home-based disabled elderly. Furthermore, 8.7% (*N* = 17) of the care recipients reported suffering from dementia.

### 3.2. Perceived Training Needs of Informal Caregivers

On average, each informal caregiver had 2.32 activities requiring training support. 86.7% (*N* = 170) of the caregivers indicated that their training needs were not being fully met and reported at least one need for targeted training activity (Figure 1). 62.7% (*N* = 123) of the caregivers identified two or more training needs. In addition, nearly 68% (*N* = 133) of the respondents reported training needs for any technical skills training activity (including self-care skills, safety supervision, functional rehabilitation, medical/nursing skills), compared to 64.3% (*N* = 126) for any intangible skills training activity (including social and emotional skills, stress management, smart device utilization).

Concerning individual training activities, the proportion of identified training needs varied significantly (Figure 2), from 19.9% (*N* = 39) for training related to “medical/nursing skills”, to 42.9% (*N* = 84) for training related to “self-care skills”. In addition to medical/nursing skills, training in other technical skills had become a widespread need for the informal caregivers. Specifically, 42.3% (*N* = 83) expressed a need for “safety supervision” training, and 40.8% (*N* = 80) felt that it was necessary to receive training in “functional rehabilitation”. Among the intangible skills, “social and emotional skills” constituted the primary need, accounting for 39.3% (*N* = 77) of the sample. Only 26.5% (*N* = 52) mentioned a perceived need for “stress management” training and 19.9% (*N* = 39) for “smart device utilization”.

### 3.3. Factors Associated with Perceived Training Needs

Table A1 reports the results of the univariate analysis, and the variables that were statistically significant with the dependent variables are included in the multivariate regression models. Table 2 reports the factors associated with different perceived training needs. Overall, the results demonstrated that caring time, care support, the individual-level characteristics of the informal caregivers, and the health status of the care recipients were significantly related to the training needs, while it differed between different kinds of training contents. Specifically, technical skills training needs were more associated with care recipients’ health status and caring-related variables, while intangible skills training needs tended to be more sensitive to the characteristics of the caregiver.

In terms of technical skills, the medical/nursing skills training need was demonstrated related to the care recipient’s ADL level and the caregiving time of the informal caregiver. The higher the care recipients’ ADL level (OR 1.443; CI 1.098, 1.896) and the informal caregiver’s caregiving time (OR 2.088; CI 1.364, 3.198), the more likely the caregivers were to have training needs for medical/nursing skills. For the self-care skills, particularly strong relationships were observed between the training need and the caregiver’s education level (OR 0.139; CI 0.033, 0.589), i.e., caregivers with an education level below elementary school were more likely to receive self-care training compared to those highly educated caregivers. In addition, a higher self-care training need was also associated with a higher care recipient’s ADL level (OR 2.359; CI 1.685, 3.303) and formal care support (OR 2.348; CI 1.071, 5.145). For the physical and functional rehabilitation training need, related factors included the care recipient’s age (OR 0.925; CI 0.885–0.966), ADL level (OR 2.390; CI 1.741, 3.280), and formal care support (OR 3.650; CI 1.578, 8.443). Specifically, informal caregivers caring for younger or higher ADL level adults were more likely to have a physical and functional rehabilitation training need. During the period of care provision, the informal caregivers who received formal care support were more likely to have this kind of training need. For safety supervision, the same correlation was found between the training need and the care recipient’s ADL level (OR 1.535; CI 1.119, 2.106), and formal care support (OR 4.084; CI 1.720, 9.697). Moreover, the informal caregivers caring for the female older adults (OR 2.554; CI 1.075, 6.069) were more inclined to receive this kind of training. A higher CCI score (OR 3.461; CI 2.148, 5.578) and caregiving time (OR 2.299; CI 1.329, 3.975) were also associated with an identified training need for safety supervision.

In terms of intangible skills, for social and emotional skills, the informal caregivers who were female (OR 2.239; CI 1.114, 4.501), had higher education (7–12 years: OR 4.343; CI 1.177, 16.022; 7–12 years: OR 11.150; CI 2.715, 45.622), who were caring for older adults with dementia (OR 3.967; CI 1.239, 12.706), and who had a higher CCI score (OR 1.588; CI 1.140, 2.212) were more likely to have a training need. For stress management skills, compared to male, married informal caregivers, those who were female (OR 3.664; CI 1.282, 10.474) or widowed/unmarried (OR 11.464; CI 3.029, 43.396) tended to be more likely to identify this training need. Furthermore, a closer relationship with the recipient (children (in law): OR 0.263; CI 0.086, 0.800; all others: OR 0.014; CI 0.001, 0.139), a higher IADL level (OR 1.880; CI 0.095, 0.780), and a longer caregiving time (OR 2.973; CI 1.509, 5.857) were also found to be positively associated with this training need. For smart device utilization, the results indicated that the informal caregiver’s age (OR 2.023; CI 1.186, 3.450) and the care recipient’s ADL level (OR 1.454; CI 1.057, 1.999) were positively correlated with higher training willingness. Notably, the squared term of age (OR 0.996; CI 0.992, 1.000), and the involvement of other caregiving support, whether formal (OR 0.185; CI 0.062, 0.551) or informal (OR 0.217; CI 0.105, 0.445), showed potential to reduce this training need.

## 4. Discussion

The results revealed that informal caregiving in Shanghai is labor intensive. On average, informal caregivers spent approximately 28 h per week providing caring activities, and over 25% of them spent more than 40 h per week. This result was markedly similar to the average weekly caregiving time spent by family caregivers in the US, UK, and Brazil, as reported in the Carer Well-being Index [34]. In the future, with the development of public payments, including medical insurance and long-term care insurance, as well as expanded access to home-based services, it is reasonable to expect that more disabled elderly people will move to home care with complex health care needs, and informal care will be exceptionally important in meeting those needs [35]. In order to alleviate the tremendous caregiving pressure and increase the sustainability of the informal care system, we strongly recommend that policy makers incorporate informal care training in a consistent and comprehensive manner in long-term care policymaking.

Our findings indicated that over 85% of the informal caregivers had a significant training need in at least one caregiving skill. The informal caregivers’ identified training need was greatest for general health-oriented activities; nearly half of the caregivers required training in self-care skills, functional rehabilitation, or safety supervision. However, the same trend did not appear in the more specialized medical/nursing skills, which differs from findings in the United States [25]. A possible reason for this is that these training needs were primarily assessed by home-health clinicians, and therefore were medically evidence-based, whereas we based our measurement on caregivers’ perceptions. Caregivers choose training content not only for its usefulness, but also for its risk, frequency, proficiency, importance, and market cost [36]. Given the lack of caregiving experience and competency, they prefer to obtain training in relatively simple, but time-consuming, care content. In contrast, specialized care tasks with high environmental and operational risks are more likely to be “contracted-out” through market purchases and public payment. In addition, this study found that stress management training was in relatively low demand across all training programs. This is an interesting finding. A possible reason is that, in a Chinese context, most informal caregivers tend to believe that “self-sacrifice” is a “natural condition” of their caregiving role, and the aim of providing better care is the core motivation for any training expectations they may have [37,38]. As a result, self-centered stress management tends to be given less importance. These findings may inform the prioritization of training resources directed to informal caregivers [39]. Existing training resources are perhaps less suited to the specific training needs of informal caregivers, as they often focus more on general chronic disease prevention literacy. In addition, given the wide range of unmet training needs, caregiving training on a larger scale is imperative. Public policymakers need to develop a skills training platform for all informal caregivers to provide them with an opportunity to connect to more training resources.

It was clear that the training needs of informal caregivers differed based on several factors. The main findings of this study were as follows:(1)Health status of the care recipient

The results revealed that the health status of the care recipient was highly associated with the perceived training needs of the informal caregivers, which again showed that the training needs of the informal caregivers were often caregiving needs oriented.

First, the range of caregivers’ training needs differed according to the state of the care recipients’ disability. Disability is usually characterized by a progressive process of loss of physical function, i.e., a gradual transition from the loss of IADL to the loss of ADL [40,41]. When the care recipient begins to exhibit a significant loss of IADL, the caregivers are more likely to experience stress in the face of unexpected caregiving tasks and role changes. Subsequently, caregivers may develop a need for stress management training, although at this point, they are often less likely to provide additional assistance [26]. As the level of disability transitions into the post-middle stage of ADL loss, caregivers are often confronted with more complex and taxing caregiving tasks. As a consequence, they are inclined to be equipped with various technical caregiving skills, including self-care, functional rehabilitation, and even medical/nursing skills. However, this “passive” transition in training needs may be inefficient. In other words, in the context of the progressive process of disability, waiting until care dependency is high before seeking skills training can be detrimental to both caregiver and care recipient [26]. Therefore, training programs should be designed for the early stage of caregiving.

Second, the level of multi-morbidity of the care recipients was also correlated with the potential training needs of the informal caregivers. A higher CCI was correlated with training needs in safety supervision and care interaction. Older adults with multi-comorbidity are more likely to suffer from sudden illness, and even death. The concern of caregivers about unexpected situations also leads to a higher need for safety supervision training, although this hypothesis is worth exploring in future work.

Third, the results also indicated that the care recipients with dementia were associated with significant social and emotional interaction training needs. This was consistent with the existing literature, which identified a great training need for the informal caregivers of older adults with dementia in how to communicate effectively [42,43,44]. Due to the language berries (e.g., word finding, naming and comprehension) and specific behavioral symptoms (e.g., memory lapses, hallucinations) of older adults with dementia, informal caregivers are often exposed to communication breakdowns while caring for them [44,45,46]. These dilemmas finally create a huge need for training in their communication skills.

Unlike the previous research that focused on the caregivers of older adults with single conditions, our study compared the differences in training needs across diseases and the severity of disability. Indeed, these results indicated that training needs were neither constant nor static [47,48]. The assessments of health status and its development/change can provide an invaluable opportunity to connect informal caregivers with specific training resources. Moreover, early training interventions become particularly important for foreseeable trends in health.

(2)Caregiving support resources

Our study confirmed that some of the training needs were found significantly related to the availability of complementary care support during the caregiving process. The informal caregivers who received any kind of assistance from formal care were more interested in training in self-care tasks, functional rehabilitation, and safety supervision, but not medical/nursing tasks training. Given the financial and time limitations of the formal care support provided by insurance and assisted care services, the intervention of formal care will likely drive the learning expectations of informal caregivers, especially for those uninterrupted, time-consuming technical caregiving tasks [25]. The involvement of formal and informal care support is also associated with less training need in smart device utilization. A plausible explanation for this is that formal/informal care support also acts as a mediator for information transfer. Caregivers are often unconsciously involved in the transmission of general knowledge, such as smart device use.

Previous research often presented informal care and formal care as alternatives to each other and tended to emphasize the important role of informal care in reducing the stress of formal care. However, our study shows that for those who are providing informal care, formal care not only provides a brief respite as a caregiving support resource, but also stimulates a desire to learn caregiving skills, which can be expected to improve their own caregiving skills and therefore, reduce the burden of caregiving. In other words, formal care and informal care are complementary. Therefore, informal caregiver training should be integrated into the framework and workflow of pertinent roles and institutions, such as family physicians and long-term care insurance. That is, informal caregiver participation and assistance should be included in the delivery of home care services. Efforts by health care systems to facilitate informal caregivers’ training could positively impact the cost and quality of system-covered home care.

(3)Caregiving stressors

As a proxy variable for physiological caregiving stressor, caregiving time was found to be significantly correlated with two technical skills training needs, i.e., medical/nursing skills and safety supervision, but not with the training need for self-care tasks nor rehabilitation skills (which were both significant in the univariate analysis). The possible explanation is that the differences in informal caregivers’ perceptions and evaluations of physiological stress, as well as the causal factors behind this stress (e.g., care demand for older adults) are the determinants of changes in informal caregivers’ training needs, whereas caregiving time as a physiological stressor is more of a mediator in overall relations. The duration of care as a source of physiological care stress, on the other hand, is more of a mediator. This hypothesis is worth exploring in future work.

As a proxy variable for emotional caregiving stressor, the relationship to the care recipient did not appear to sway most of the skills training needs of the informal caregivers. However, it is worth noting that this emotional stressor was positively associated with the stress management training need. Intimate relationships tended to be imbued with more social norms and altruistic principles [49], and were resilient to external circumstances. Therefore, caregivers with more intimate caregiving relationships are more likely to have strong perceptions of caregiving responsibilities, and often have difficulty coping with their loved one’s suffering from health damage [50], and are therefore more likely to suffer from a significant psychological burden. When these burdens cannot be resolved, informal caregivers will naturally develop a strong stress management training need. Consequently, close attention should be paid to the need for timely psychological adjustment for those caregivers who have an intimate relationship with the care recipient.

(4)Personal attributes of caregiver

The personal attributes of informal caregivers, including gender, age, education level, and marital status, were significantly correlated with their potential training needs. Moreover, compared to the technical skills training need, caregivers’ flexible, intangible skills training needs were more likely to fluctuate with changes in the caregivers’ personal attributes. For example, in line with the existing research, female informal caregivers reported greater training needs in the areas of social and emotional skills and stress management skills. Interestingly, in our study, the caregivers with a lower education level placed a higher value on training in self-care tasks, but a lower value on training in social and emotional skills. This finding is also not inconsistent with recent research findings, as education may have a lasting and formative impact on an individual’s thinking perceptions and emotional life [51]. Therefore, a comprehensive caregiver assessment is expected to constitute an especially valuable tool in providing adaptive training guidance and support.

In aggregate, our study suggests that a “one-size-fits-all” approach is not applicable to informal care training [20]. However, unified training content is currently provided by third parties in the delivery practice of informal care training in Shanghai. Thorough investigation of informal caregivers, as well as care recipients, is critical to the development of an appropriate and effective training program [52]. Therefore, we believe that the setting of training programs should be tailored and individualized based on the comprehensive assessment of care recipients’ health condition, care acceptance, caregiver characteristics, and subjective needs. Furthermore, training programs should also be designed for the early stages of caregiving and include follow-up training sessions that are tailored (individual level) or relevant (group level) through the progression caregiving.

### 4.1. Limitations

This study is subject to several limitations. First, our sample was informal caregivers who participated in the “The Older Adult Helping Program” in Shanghai, which may not fully represent all the caregivers in China. However, since the program was open to all informal caregivers providing care to elderly who need assistance in their homes, it is acceptable to some extent. Second, due to the difficulties in gaining access to informal caregivers and their lack of awareness of training, our effective sample size was relatively small, which may have affected the validity of the results and limited further causal analyses. Third, our data were collected during the pandemic, so our findings may be relevant to the studied period. Specifically, the impact of the pandemic on the health status of older adults, and on the caregiving burden of informal caregivers, may result in changes in perceived training needs. Fourth, we investigated the disability and illness status of the care recipients in detail to further analyze the potential correlation between these factors and the training needs of the informal caregivers. However, since this study aimed to elucidate the training needs of informal caregivers in general, the state of elders’ particular situation was not surveyed, i.e., whether they were in a sudden onset of illness or in a chronic condition. Pertinent literature has identified significant associations between the care transitions surrounding acute health events and the more specific training needs of caregivers. Finally, we adopted a self-reported approach to determine the perceived training needs of informal caregivers, and caregivers may or may not be aware of services that could benefit them. Future research could develop a hybrid evaluation method that combines qualitative and quantitative methods to precisely identify the training needs of informal caregivers.

### 4.2. Practice Implications

Appropriate skills training should be provided for informal caregivers to alleviate huge care pressure and improve their health. However, there is limited information about caregivers’ perceived training needs in China. The present study highlights the importance of informal caregivers’ involvement in constructing training support strategies and confirms their perceived training needs from a caregiver perspective. The findings suggest the formation of personalized training programs based on informal caregiver characteristics and factors, focusing on their preferences, priorities, and dynamic changes. The results also indicated that training should be incorporated into the workflow of formal care. These views are expected to provide valuable guidance for improving informal caregiving support and health outcomes in China.

## 5. Conclusions

This paper explored the perceived training needs of informal caregivers and their associated factors in Shanghai. The findings provided an overview of the training needs of the caregivers and the comparable results for international research in this area. This study revealed that most informal caregivers have unmet training needs. General health and functional activities, as well as social and emotional skills, were the core training concerns of the caregivers. In addition, our study provides interesting insights into the differential impact of care burden and care support on different perceived training needs; it was found that complementary formal care support was associated with training needs, in addition to the personal characteristics of caregivers and care recipients. A standardized contextual assessment and individualized training programs are, therefore, critical for the provision of effective support to informal caregivers. Moreover, it is necessary to integrate informal care into the formal care framework and to provide active policies to support the synergistic development of the two.

## Figures and Tables

**Figure 1 healthcare-12-02369-f001:**
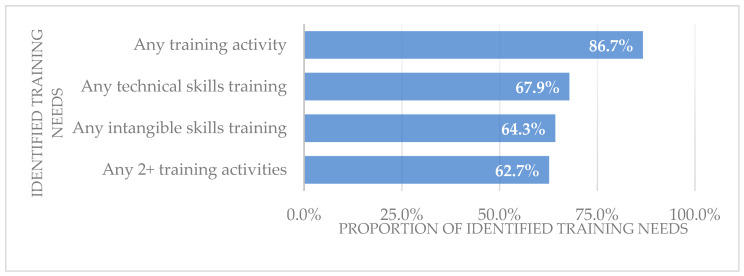
Rate of different types of training needs among informal caregivers.

**Figure 2 healthcare-12-02369-f002:**
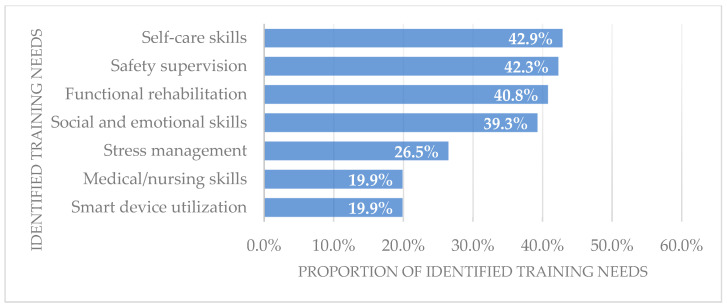
Rate of each identified training need among informal caregivers.

**Table 1 healthcare-12-02369-t001:** Demographic characteristics of informal caregivers and care recipients.

Variables	Informal Caregivers		Care Recipients	
	N/(Mean)	%/(SD)	N/(Mean)	%/(SD)
Individual-level characteristics				
Gender				
Male	66	33.7	74	37.8
Female	130	66.3	122	62.2
Age	(61.18)	(13.16)	(78.76)	(9.847)
Education level				
<7 years	21	10.7	74	37.8
7–12 years	128	65.3	103	52.6
>12 years	47	24.0	19	9.7
Marital status				
Married	161	82.1	92	46.9
Widowed	8	4.1	87	44.4
All others	27	13.8	17	8.7
Income (CNY/monthly)				
<4000	98	50.0	124	63.3
4000–6000	66	33.7	50	25.5
>6000	32	16.3	22	11.2
Work status				
Employed	50	25.5		
Retired/unemployed	146	74.5		
Living arrangement				
Co-residence	133	67.9		
Live apart	63	32.1		
ADL			(2.02)	(1.614)
IADL			(4.37)	(1.709)
CCI			(1.72)	(0.981)
Dementia				
Yes			17	8.7
No			179	91.3
Caregiving stressors				
Caregiving time	(28.02)	(19.215)		
Relationship to the recipient				
Spouse	59	30.1		
Children (in law)	102	52.0		
All others	35	17.9		
Caregiving supports				
Formal care support				
Yes	102	52.0		
No	94	48.0		
Other informal care support				
Yes	171	87.2		
No	25	12.8		

**Table 2 healthcare-12-02369-t002:** Factors associated with perceived training needs of informal caregivers.

	Technical Skills Training Need				Intangible Skills Training Need		
	Medical/ Nursing Skills	Self-Care Skills	Physical and Functional Rehabilitation	Safety Supervision	Social and Emotional Skills	Stress Management	Smart Devices Utilization
	OR (95% CI)	OR (95% CI)	OR (95% CI)	OR (95% CI)	OR (95% CI)	OR (95% CI)	OR (95% CI)
Characteristic of informal caregiver							
Gender (man = ref)					2.239 ** (1.114, 4.501)	3.664 ** (1.282, 10.477)	
Age							2.023 ** (1.186, 3.450)
Age^2^							0.996 * (0.992, 1.000)
Education (<7 year = ref)							
7–12 years		0.447 (0.122, 1.634)			4.343 ** (1.177, 16.022)		
>12 years		0.139 ** (0.033, 0.589)			11.150 ** (2.725, 45.622)		
Marital (others = ref)						11.464 *** (3.029, 43.396)	
Characteristics of care recipient							
Gender (man = ref)				2.554 ** (1.075, 6.069)			
Age			0.925 *** (0.885, 0.966)				
ADL	1.443 ** (1.098, 1.896)	2.359 *** (1.685, 3.303)	2.390 *** (1.741, 3.280)	1.535 ** (1.119, 2.106)			1.454 * (1.057, 1.999)
IADL						1.880 ** (1.314, 2.689)	
CCI				3.461 *** (2.148, 5.578)	1.588 ** (1.140, 2.212)		
Dementia (no = ref)					3.967 * (1.239, 12.706)		
Caregiving stressors							
Caregiving time	2.088 *** (1.364, 3.198)			2.299 ** (1.329, 3.975)		2.973 ** (1.509, 5.857)	
Relationship to care recipient (spouse = ref)							
Children (in law)						0.263 * (0.086, 0.800)	
All others						0.014 *** (0.001, 0.139)	
Caregiving supports							
Formal care (no = ref)		2.348 ** (1.071, 5.145)	3.650 ** (1.578, 8.443)	4.084 ** (1.720, 9.697)			0.185 ** (0.062, 0.551)
Other informal care (no = ref)							0.185 ** (0.062, 0.551)
Constant	0.086 ***	0.213 **	24.636 **	0.008 ***	0.030 ***	0.010 ***	0.003 ***
R^2^	0.188	0.352	0.314	0.427	0.151	0.400	0.274
Adjust R^2^	0.297	0.473	0.424	0.574	0.204	0.583	0.434

Age^2^ denotes the squared term of the Age variable. * *p* < 0.10, ** *p*< 0.05, *** *p* < 0.001.

## Data Availability

Data are available upon reasonable request by contacting the corresponding author.

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
