# Peer review of "Perceived Training Needs of the Informal Caregivers of Older Adults: A Cross-Sectional Study"

_healthcare, 2024, doi:10.3390/healthcare12232369_

Round 1
Reviewer 1 Report
Comments and Suggestions for Authors
Given the ageing population and increasing burden on caregivers, the research on caregivers’ needs is relevant and has potential to address some of the gaps in the current knowledge on the subject. However, it is recommended for the authors to review the Methods and Results sections and address the following comments:
Lines 188-189
The threshold for a p value less than 0.200 was adopted in the univariate analyses (Table A1). I’m not convinced whether such high threshold can be applied to test a null hypothesis.
Line 99: I would recommend rewording to avoid repetitive words, for example “inclusion criteria were…”
Lines 99-100: Could you define what the “primary informal caregiver” role is?
Line 107: “2.24% (n = 6) were excluded for excessive work hours (more than 16 hours a day)”. Excessive care hours wasn’t mentioned in the inclusion criteria, so a bit unclear what the authors referred to.
Lines 128-129 It would be useful to have some examples of activities for ADL and IADL scales, explain how these activities are scored and how scores can be interpreted.
153-154 “Two rounds of expert consultations and a pilot survey of 50 informal caregivers were further conducted to adapt to care practices of informal caregivers in Shanghai”. It would be good to give more detail on the described consultations, for example who the experts were, what questions and topics were covered in consultations and the survey.
162-163 A bit unclear what the following means: “…and household chores was eliminated for no caregiver reported”.
Table 2: no need to have both 95% CI and p value, only reporting Cis is enough. Also, no need to present both β coefficient(SE) and odds ratios (95% CI). However, having the number of
Titles of all tables: avoid using “analysis” in the titles, as the tables present the results. For example, Table 1 could be named “Demographic characteristics of informal caregivers, care receivers, and caregiving related measures”. Table 2 could be “Factors associated with medical-oriented technical skill training needs”.
Author Response
Response to Reviewer 1 Comments
Thank you for providing us with the valuable opportunity to modify our manuscript and resubmit it. Your suggestions and comments are very helpful. We have greatly improved the manuscript according to your important and enlightening suggestions. We hope that the manuscript is now ready for publication.
Point 1:
Lines 188-189: The threshold for a p value less than 0.200 was adopted in the univariate analyses (Table A1). I’m not convinced whether such high threshold can be applied to test a null hypothesis.
Response 1:
Thank you for this insightful question.
As you note, the selection parameter (that is, the significance level to decide whether an effect should be retained in the model) must be carefully chosen when screening candidate predictor variables through univariate analysis. On this issue, most researchers tend to believe that too strict p-value settings are likely to lead to the omission of important adjustment variables that need to be controlled in the multivariate model. Smaller values such as 0.05 or 0.01 are only recommended for very large sample sizes (EPV of 100 or above). In the vast majority of applications, especially when the regression model does not have many predictor variables and the sample size is not that large, a value of 0.20 or 0.157 (corresponding to selection based on AIC) or even 0.5 (resulting in very mild selection) will be a better choice (Heinze G, Dunkler D, 2017).
Therefore, we established a more moderate parameter selection (p < 0.20) in the univariate analysis. Interestingly, it is found that some predictor variables did not pass the significant level test of p < 0.05 in the univariate analysis, but showed a stronger correlation with the core explanatory variables in the multivariate analysis (p < 0.05).
We further fully explained the significance level setting for the selection of predictor variables through univariate analysis in the method section. Please see the changes in the revised manuscript.
Reference:
Heinze G, Dunkler D. Five myths about variable selection. Transplant International. 2017;30(1):6-10. doi:10.1111/tri.12895
Point 2:
Line 99: I would recommend rewording to avoid repetitive words, for example “inclusion criteria were…”
Response 2:
Thank you for this useful suggestion. We have reworded the sentence as you suggested, and the whole paragraph is now more concise and readable. Please see changes in the revised manuscript.
Point 3:
Lines 99-100: Could you define what the “primary informal caregiver” role is?
Response 3:
Thank you for this important suggestion.
The primary informal caregivers should be those who “have spent at least 1 hour per day performing care tasks or provided care consistently for at least 1 month (a minimum of 1 time per week during that month)”. These were mentioned in detail in the respondent inclusion criteria.
This was mainly to avoid some family members living with the disabled older adults (who may actually provide little or no care) being defaulted to informal caregivers, which may result in statistical bias (those who have only transient experiences of informal caregiving may be unaware of what they need in terms of training).
We realized that the presentation of the participants screening criteria was not clear. Therefore, we further fully explained the criteria by numbering them one by one. Please see the changes in the revised manuscript.
Point 4:
Line 107: “2.24% (n = 6) were excluded for excessive work hours (more than 16 hours a day)”. Excessive care hours wasn’t mentioned in the inclusion criteria, so a bit unclear what the authors referred to.
Response 4:
Thank you for pointing this out to us.
Prior to the questionnaire, we set a series of inclusion criteria for participants, which primarily attempted to filter out participants at the questionnaire stage whom we would not include in the study.
Further data cleaning was performed to improve the quality of the sample data and the accuracy of the analysis. And we found that some participants reported excessive caregiving time, especially when they live with the care recipient (some of them intended to report 24 h/d of caring), which was obviously exaggerated. Therefore, we excluded caregivers who reported to have a caregiving time over 16 h/d to avoid over-pledging bias (then there will be less than 8 h/d for rest).
Point 5:
Lines 128-129: It would be useful to have some examples of activities for ADL and IADL scales, explain how these activities are scored and how scores can be interpreted.
Response 5:
Thank you for this valuable suggestion.
The ADL scale consists of 6 items centered primarily on basic physical needs: eating, dressing, continence, toileting, bathing and indoor ambulating. The IADL scale consists of 8 more complex activities related to independent living in the community, including the ability to use telephone, shopping, food preparation, housekeeping, laundry, mode of transportation, responsibility for own medications, ability to handle finances. Informal caregivers were asked whether the care recipient they cared for did (1) or did not (0) have any difficulty with these activities. All item scores were summed separately to obtain the ADL (scale from 0 to 6; Cronbach’s alpha 0.744) and IADL scores (scale from 0 to 8; Cronbach’s alpha 0.753). A higher score of ADL/IADL usually means a higher level of disability.
The composition and measurement of the ADL and IADL scales were detailed in the materials and methods section. Please see the changes in the revised manuscript.
Point 6:
Lines 153-154 “Two rounds of expert consultations and a pilot survey of 50 informal caregivers were further conducted to adapt to care practices of informal caregivers in Shanghai”. It would be good to give more detail on the described consultations, for example who the experts were, what questions and topics were covered in consultations and the survey.
Response 6:
Thank you for this valuable suggestion.
A 6-member consulting team (including three trainers from informal care training institutions and three experts in the field of long-term care) first translated, localized and revised the training content based on their professional knowledge and practical experience. And a training framework with seven categories was initially developed, which include medical/nursing skills, self-care skills, functional rehabilitation, safety supervision, household chores, social/emotional skills, and stress management. Thereafter, a further pilot survey based on 50 informal caregivers was conducted to verify the reliability of the content. And an open box was provided where participants were asked to report any other self-identified training needs. Interestingly, in the pilot survey, no participant reported “household chores” skills as a training need; instead, “smart devices utilization” skills training was repeatedly mentioned in the open box. Therefore, in the second round of consultation, all experts reached a consensus to replace “household chores” skills with “smart devices utilization” skills in the training need option.
According to your suggestion, we have added these consultation details in the materials and methods section, please see changes in the revised manuscript.
Point 7:
Lines 162-163 A bit unclear what the following means: “…and household chores was eliminated for no caregiver reported”.
Response 7:
Thank you for pointing this out to us.
In the pilot survey, the “household chores” skills included in the original categorization of training needs was not mentioned by any informal caregivers. Instead, “smart devices utilization” skills training was repeatedly mentioned in the open box. Therefore, in the second round of consultation, all experts reached a consensus to replace “household chores” skills with “smart devices utilization” skills in the training need option.
Considering the confusion caused by the current improper phrasing, we have improved our expression in the revised manuscript. Please see the changes in the revised manuscript.
Point 8:
Table 2: no need to have both 95% CI and p value, only reporting Cis is enough. Also, no need to present both β coefficient(SE) and odds ratios (95% CI). However, having the number of Titles of all tables: avoid using “analysis” in the titles, as the tables present the results. For example, Table 1 could be named “Demographic characteristics of informal caregivers, care receivers, and caregiving related measures”. Table 2 could be “Factors associated with medical-oriented technical skill training needs”.
Response 8:
Thank you for these important suggestions.
We have carefully considered your suggestions and have made corresponding revisions. Please see the changes in the revised manuscript.

Reviewer 2 Report
Comments and Suggestions for Authors
IRB approval is mandatory.
The article is well-written, but the methodology is not clearly presented.
The items of questionnaires are not indicated.
The exclusion of participants is unclear.
The conclusion is poor.
The study does not add any new elements.
Author Response
Response to Reviewer 2 Comments and Suggestions
Point 1:
IRB approval is mandatory.
Response 1:
Thank you for your important suggestion.
Since the college of Philosophy, Law and Political Science, Shanghai Normal University has not established an ethical review institution, we had no way to conduct ethical review for our study.
However, we still conducted a design justification and applied for permission to administer the questionnaire to the academic committee of the college before the study began. The committee agreed our research, and concluded that our research only involves the study of participants' behavior with no risk or minimal risk, and all of the interviewed caregivers are de-identified, which means the project does not involve any ethical issue.
Meanwhile, in order to avoid raising possible ethical risks, we also adopted informed consent and advance notification before the survey so as to avoid any possible harm to the respondents.
We have tried our best to keep our research free of ethical risks, and we appreciate your understanding.
Point 2:
The article is well-written, but the methodology is not clearly presented.
Response 2:
Thank you for your valuable comment.
In response to your suggestions, we have made extensive revisions to the materials and methods subsections.
First, we have further detailed our subject recruitment process, especially the whole process of screening the sample. And, we have also added the sample size calculation. Second, we have elaborated the whole process of conducting the questionnaire. This includes information on who helped, where and when the interview conducted, why FTF interviews were chosen, etc. Third, we have explained in detail the selection and operationalization of the research variables. In particular, for the dependent variable, perceived training needs, we have provided details about how the training needs categories were obtained and how the variables were operationalized. Finally, we have provided further justification of the choice of methodology for the statistical analysis of the data.
Thank you very much for your comments and suggestions to assist us in improving the clarity and readability of the methods section. We hope that our changes can be recognized by you.
Point 3:
The items of questionnaires are not indicated.
Response 3:
Thank you for your valuable comment.
A questionnaire was developed and designed to obtain data, with information including sections on sociodemographic information about informal caregivers and care recipients, caregiving-related information, and perceived training needs content.
Based on your suggestions, we have briefly described the content of several boards involved in the questionnaire in the method section. Please see the change in the revised manuscript.
Point 4:
The exclusion of participants is unclear.
Response 4:
Thank you for your valuable comment.
Considering the differences in the populations cared for by informal caregivers and the cognitive ability to answer questions related to caregiving training needs, several requirements were established for inclusion of the respondents. The informal caregiver should be: (a) be 16-year-old or older; (b) be the primary caregiver, who have spent at least 1 hour per day performing care tasks, or provided care consistently for at least 1 month (a minimum of 1 time per week during that month); (c) caring for older adults with any ADL/IADL impairment; (d) not been registered, or financially compensated by any formal for-profit institutions. 37 informal caregivers were filtered out based on the inclusion criteria, among them, 2 for younger than 16-year-old, 9 for not primary caregivers, 4 for had professional credentials or were market-employed nannies, and 12 for no longer had an ADL/IADL impairment of care recipient. And finally, a total of 268 eligible informal caregivers voluntarily participated in the survey.
Details of the inclusion/exclusion criteria for participants have been added in the method section, please see the change in the revised manuscript.
Point 5:
The conclusion is poor.
Response 5:
Thank you for your valuable comment.
We have further enriched the conclusions section.
For example, we have added an extended discussion of the relationship between formal care support resources and the perceived training needs of informal caregivers. Previous research often presented informal care and formal care as alternatives to each other and tended to emphasize the important role of informal care in reducing the stress of formal care. However, our study shows that for those who are providing informal care, formal care not only provides a brief respite as a caregiving support resource, but also stimulates a desire to learn caregiving skills, which can be expected to improve their own caregiving skills and therefore, reduce the burden of caregiving. In other words, formal (professional) care and informal care are complementary, implying that an extensive formal support framework provides an environment where both informal and formal caregivers function optimally.
Please see the changes in the revised manuscript.
Point 6:
The study does not add any new elements.
Response 6:
Thank you for your valuable comment.
Our study is innovative in two areas.
First, this study provides, to the best of the authors’ knowledge, the first empirical evidence of the perceived training needs of informal caregivers in Shanghai, China. In general, the preferences for training needs of informal caregivers and the influencing factors vary from case to case. These include demographic aspects (e.g. average age of the population), economic and cultural aspects (e.g. caregiving is a family responsibility) and the current form of care in the country. In order to have a reliable understanding of preferences and training needs, each country and even each region must be considered individually. And this study provides an empirical experience that can be used to inform informal care training support in China.
Second, our research revealed a significant correlation between formal caregiving support and caregivers' specific caregiving skills training needs. This was not found in previous studies. This means that for those who are providing informal care, formal care not only provides a brief respite as a caregiving support resource, but also stimulates a desire to learn caregiving skills, which can be expected to improve their own caregiving skills and therefore, reduce the burden of caregiving. Therefore, the findings of this study provide empirical evidence for policy initiatives to meet the training needs of informal caregivers and reduce the cost of long-term care through home-based care services and training visits by professional formal caregivers.

Reviewer 3 Report
Comments and Suggestions for Authors
The study attempts to understand caregiver needs and outline the factors associated with their mentioning.
Although this study is interesting, it has many pitfalls that prevent its publishing in a current form. I first outline my major concerns with this paper and then proceed to additional feedback.
My first major concern was about the literature review. The study is self-encapsulated and does not provide an extensive outline of the previous research. Therefore, the question of whether this study is necessary, important, and/or valuable remains open. The authors are required to review the relevant literature more thoroughly. These are the articles that I would like to ask the authors to incorporate in the manuscript. Please also locate others.
Bom, J., Bakx, P., Schut, F., & Van Doorslaer, E. (2019). The impact of informal caregiving for older adults on the health of various types of caregivers: a systematic review. The Gerontologist, 59(5), e629-e642.
Plöthner, M., Schmidt, K., De Jong, L., Zeidler, J., & Damm, K. (2019). Needs and preferences of informal caregivers regarding outpatient care for the elderly: a systematic literature review. BMC geriatrics, 19, 1-22.
Verbakel, E., Metzelthin, S. F., & Kempen, G. I. (2018). Caregiving to older adults: Determinants of informal caregivers’ subjective well-being and formal and informal support as alleviating conditions. The Journals of Gerontology: Series B, 73(6), 1099-1111.
My second major concern refers to the choice and the performance of the statistical analyses. The tables contain very confusing, inconclusive, and sometimes misleading information thereby making it difficult to compare between the examined models. The choice of including only those variables that exhibited significant associations at the univariate level seems missing the target. Moreover, I am generally not sure that there is a need for the univariate analysis. The authors are therefore required to perform the multivariate analysis again and subsequently redesign the tables. I would also like the authors to address the way the dependent variables were operationalized. Specifically, what were the reference categories in each dependent variable examined?
Now I proceed to additional feedback. But beforehand, I would like to ask the authors to go through the article and replace the words "elderly", "senior" and similar terms with "older adults". This includes the manuscript title.
ABSTRACT
1. "With the rapid expansion of the aging population..." what does this mean? Please rephrase.
2. "in Shanghai." Only there? Unclear why you decided to mention only this city.
3. "However, caregivers’ perceived training needs have not been examined recently". They have been eventually. Please either remove or rephrase this sentence as it currently signals poor review of the literature.
4. "well-suited intervention programs" aimed at what? Please add.
5. "The aims of this 20 research were to explore areas in which informal caregivers perceived that they would benefit from 21 further training, and to identify variables associated with caregiver training needs in Shanghai, China". Please amend this sentence as it sounds unclear.
6. "eligible" in which terms? Please add.
7. "Univariate and multivariate analysis" Which ones? Please mention instead.
8. "Most of the informal caregivers had at least one significant training need". How much is "most"? Moreover, how many had two or more needs? Please add all this information.
9. "Factors associated with technical skills training needs were primarily care recipients’ diseases/conditions and caring-related variables." A very vague sentence. Please give examples.
INTRODUCTION
1. "Influenced by intergenerational compensation and traditional family culture, informal care mainly provided by family members has long been the primary way to meet the care needs of the elderly in China". Please elaborate better on this issue as it provides a motivation for your study.
2. "With the development of China's rapidly expanding elderly population" - what does this collection of words mean? Please rephrase.
3. Line 46 - please replace the word "preparation" with "training".
4. "When these high-intensity caregiving tasks are overlain with significant caregiving skill gaps, caregivers can experience substantial stress and markedly decrease quality-of-care for the recipients". What is the mechanism? Why would people experience stress and burden? What does caregiver stress model maintain about that? Please discuss all this.
5. "In China, however, informal caregivers contribute the bulk of long-term care services, but with little access to supportive services for themselves." But what are the existing nationwide intiatives? Please outline.
6. The entire section provides absolutely no relation to the previous research in the field. Why would the scientific community need this study? What does it discover that the others did not succeed in? Please elaborate on these issues using the articles I cited above and the other ones.
7. Who is going to benefit from the study findings and in which ways? Please dedicate a paragraph for this.
MATERIALS AND METHODS
1. "The fourth iteration of the “Elderly Helping Program” was implemented in 2021," When were the three first iterations held? Please add.
2. "...a set of relevant conditions," What were they? Please add.
3. Line 99 - Why the 16-year bound was chosen as the lowest limit? Please justify in the text.
4. "the primary informal caregiver". How was this identified? Please add.
5. "with any physical activity impairment" How did you define it? Please add.
6. How was the "consistently" (line 101) defined? Please add.
7. "A total of 268 eligible informal caregivers participated in the survey." Out of how many approached/filtered out? Please provide a full flow of the participants, also using a figure.
8. "Face-to-face questionnaire-based surveys were administered during December 3, 2021 to January 23, 2022.". In which settings? How was the entire process of interviewing organized? Why it was decided to stop the data collection on the second date mentioned? Why FTF interviews were chosen? Please add all this information. Please also replace the words "during... to" with "between.... and ...."
9. Were the education and the income variables used in a continuous form or were dichotomized? Please better elaborate here. If the former is right, please explain why you treated the three-category variable as such.
10. How were the other variables defined for the purposes of the regression analysis? Please add the detailed information.
11. I do not understand what the "hierarchical relationship" (line 130) means. Please rephrase.
RESULTS
1. Table 1 - it is unclear what the statistics for the need variables mean. Please show clear and comprehendable measures.
2. Please divide Figure 1 into two separate ones as there is currently a lot of information inside. Please also arrange the figures in some order (either descending or ascending).
3. The reason for the separation between the tables is unclear. If you still decide to do this after reexamination of the models, as was asked above, please justify this separation.
LIMITATIONS
There are two meaningful limitations that were not mentioned. First, a relatively small sample. Second, the potential relevance of the findings for the studied period only. Note that the data was collected during the pandemic.
Comments on the Quality of English LanguageA quite poor one. There are too many awkwardly or incorrectly sounding parts of sentences (e.g. "..does not take informal caregivers' demand substantially into consideration." or "...well-suited intervention programs."), irrelevant or misused words and expressions. A thorough proofreading is essential.
Author Response
Response to Reviewer 3 Comments
The study attempts to understand caregiver needs and outline the factors associated with their mentioning. Although this study is interesting, it has many pitfalls that prevent its publishing in a current form. I first outline my major concerns with this paper and then proceed to additional feedback.
Thank you for providing us with the valuable opportunity to revise our manuscript and resubmit it. We improved our manuscript greatly according to your important and enlightening suggestions. We not only revised the content of the manuscript, but also improved the readability and clarity of the manuscript according to your suggestions. We hope that the manuscript is now ready for publication.
Major Concerns with This Paper:
Point 1:
My first major concern was about the literature review. The study is self-encapsulated and does not provide an extensive outline of the previous research. Therefore, the question of whether this study is necessary, important, and/or valuable remains open. The authors are required to review the relevant literature more thoroughly. These are the articles that I would like to ask the authors to incorporate in the manuscript. Please also locate others.
Bom, J., Bakx, P., Schut, F., & Van Doorslaer, E. (2019). The impact of informal caregiving for older adults on the health of various types of caregivers: a systematic review. The Gerontologist, 59(5), e629-e642.
Plöthner, M., Schmidt, K., De Jong, L., Zeidler, J., & Damm, K. (2019). Needs and preferences of informal caregivers regarding outpatient care for the elderly: a systematic literature review. BMC geriatrics, 19, 1-22.
Verbakel, E., Metzelthin, S. F., & Kempen, G. I. (2018). Caregiving to older adults: Determinants of informal caregivers’ subjective well-being and formal and informal support as alleviating conditions. The Journals of Gerontology: Series B, 73(6), 1099-1111.
Response 1:
Thank you for this enlightening suggestion.
We first added a literature review on caregiving stress and training need for informal caregivers in the introduction section, which included the recommended papers. Following this, we further emphasized the importance and research value of this study in the introduction section, based on the gaps in existing research.
We thank you sincerely. And we hope that our revision can largely improve the issues you have raised. Please see the detailed changes in the revised manuscript.
Point 2:
My second major concern refers to the choice and the performance of the statistical analyses. The tables contain very confusing, inconclusive, and sometimes misleading information thereby making it difficult to compare between the examined models. The choice of including only those variables that exhibited significant associations at the univariate level seems missing the target. Moreover, I am generally not sure that there is a need for the univariate analysis. The authors are therefore required to perform the multivariate analysis again and subsequently redesign the tables.
Response 2:
The purpose of this study was to explore the factors associated with the perceived training needs of informal caregivers. When the number of candidate predictor variables considered or known confounders is large, a “full” model including all candidate predictors as explanatory variables is often considered impractical for small samples of data use or, in the extreme case, may even lead to a failure of the construction (Heinze G, Dunkler D, 2017). Therefore, variable selection approaches are often employed, mostly based on evaluating P-values for testing regression coefficients against zero. For example, Martinez-Selles et al. (2015) used univariate screening of effects to build a multivariable model for survival after heart transplantation. After univariate selection, Rodriguez-Pera lvarez et al. (2015) employed “backward elimination” which means that first a multivariable model was built with all predictors selected by univariate screening in a first step. Then, nonsignificant predictor variables were sequentially eliminated and models re-estimated until all variables remaining in the model show significant association with the outcome. Therefore, our article draws on these studies by building a multivariate model with all predictors screened by univariate analysis. In addition, considering the overly strict p-value setting for the univariate analysis may lead to the omission of important adjustment variables that need to be included in the model. A more moderate parameter selection (p<0.2) was set for the variables selected in the univariate analysis.
Following your suggestion, we have attempted to include all predictor variables to construct a “full” model. However, it appears that the regression results of this approach do not differ significantly from the existing results. In particular, the significance of the regression coefficients between the variables we focused on (e.g., caregiving support, caregiving burden, etc.) and the training needs did not change. Therefore, we decided to retain the existing statistical analysis approach.
In response to your question “The tables contain very confusing, inconclusive”. We have redesigned the table to make it clearer.
Thank you very much for your suggestions, and we hope that our interpretations and modifications will further improve the performance of the study.
References:
- Heinze G, Dunkler D. Five myths about variable selection. Transplant International. 2017;30(1):6-10. doi:10.1111/tri.12895
- Reed RM, Eberlein M. Donor/recipient sex mismatch and survival after heart transplantation: only an issue in female recipients? Transpl Int. 2015;28(5):622-622. doi:10.1111/tri.12502
- Rodriguez-Peralvarez M, Garcia-Caparros C, Tsochatzis E, et al. Lack of agreement for defining “clinical suspicion of rejection” in liver transplantation: a model to select candidates for liver biopsy. Transpl Int. 2015;28(4):455-464. doi:10.1111/tri.12514
Point 3:
I would also like the authors to address the way the dependent variables were operationalized. Specifically, what were the reference categories in each dependent variable examined?
Response 3:
Thank you for this valuable suggestion.
The dependent variables in this paper are the perceived training needs of each training activity. Perceived training needs were obtained via direct questioning. The question posed was as follow:
Which of the following training activities would you consider necessary to be trained so as to help better manage daily caregiving tasks?
A training need was considered to exist when any of the subcategory items of a training activity were necessary to provide the training. Each training activity was a dichotomous variable, with 1 indicating “yes” and 0 indicating “no.”
In response to your helpful suggestion, we have added the section on how to operationalized the dependent variables. Please see the change in the revised manuscript.
Point 4:
I would like to ask the authors to go through the article and replace the words "elderly", "senior" and similar terms with "older adults". This includes the manuscript title.
Response 4:
Thank you for this important suggestion.
In response to your recommendation, we rechecked the entire article and replaced the words “elderly”, “senior” and similar terms with “old adults”. We hope that the improvement will further increase the readability of the manuscript.
ABSTRACT
Point 5:
"With the rapid expansion of the aging population..." what does this mean? Please rephrase.
Response 5:
Thank you for pointing it out to us.
The point we're trying to express is that “as the aging population continues to grow…”.
We have fixed this sentence, please see the change in the revised manuscript.
Point 6:
"in Shanghai." Only there? Unclear why you decided to mention only this city.
Response 6:
Thank you for your insightful comment.
Only Shanghai was mentioned mainly because this study was conducted with informal caregivers in Shanghai, China.
In China, informal caregivers have long been viewed as an established resource and have not been given focused attention as key stakeholders. Specifically, nationwide advocacy has been inclined to indirectly support informal caregivers by providing financial and service assistance to the care recipients. This includes the provision of living allowances, home care services, and long-term care insurance for older adults. Policy initiatives targeting informal caregivers remain limited, such as individual income tax deductions for children supporting their parents, and nursing leave for one-child families. Supportive policies such as counseling and training for informal caregivers are unavailable. In recent years, some cities in China have begun to explore informal caregiver-oriented caregiving training support programs, such as Shanghai, Shenzhen, and Beijing. Among them, Shanghai is the earliest city to do so. Therefore, this paper expects to summarize the problems and experiences encountered in the implementation of related policies through the study of informal care training needs in Shanghai, so as to provide a reference for other regions in China.
However, in terms of the research background, the huge demand for caregiving due to population aging is a common global trend, and the provision of appropriate training support for informal caregivers has also become an important caregiver support strategy in most countries/regions. Therefore, based on your suggestion, we decided to contextualize the study within a larger framework. We have modified both the abstract and introduction section. Please see the changes in the revised manuscript for more details.
Point 7:
"However, caregivers’ perceived training needs have not been examined recently". They have been eventually. Please either remove or rephrase this sentence as it currently signals poor review of the literature.
Response 7:
Thank you for this important suggestion.
Actually, what we are trying to express is that “research on the training needs of informal caregivers in China is still insufficient, with existing studies focusing on caregiver burden”.
Following your suggestion, we have removed this sentence. In addition, we have added a literature review on the training needs of informal caregivers in the introduction section. Please see the changes in the revised manuscript.
Point 8:
"well-suited intervention programs" aimed at what? Please add.
Response 8:
Thank you for this important question.
Effective intervention programs aimed at reducing the caregiving burden of informal caregivers and avoiding the deterioration of heath associated with caregiving.
We have added these contents. Please see the change in the revised manuscript.
Point 9:
"The aims of this research were to explore areas in which informal caregivers perceived that they would benefit from further training, and to identify variables associated with caregiver training needs in Shanghai, China". Please amend this sentence as it sounds unclear.
Response 9:
Thank you for pointing this out to us.
We have fixed this sentence, please see the change in the revised manuscript.
Point 10:
"eligible" in which terms? Please add.
Response 10:
Thank you for this important suggestion.
A number of inclusion criteria were established to ensure that the survey included eligible respondents. The informal caregiver should be: (a) be 16-year-old or older; (b) be the primary care-giver, who have spent at least 1 hour per day performing care tasks, or provided care consistently for at least 1 month (a minimum of 1 time per week during that month); (c) caring for older adults with any ADL/IADL impairment; (d) not been registered, or financially compensated by any formal for-profit institutions.
The above has been described in the methods section. Considering the word limit, we will not describe it in detail in the abstract section.
Point 11:
"Univariate and multivariate analysis" Which ones? Please mention instead.
Response 11:
Thank you for pointing this out to us.
We primarily adopted multivariate analysis to explore the factors associated with informal caregivers’ perceived training needs, while univariate analysis was employed to screen possible predictor variables for the multivariate analysis.
We have modified the phrasing of this sentence. Please see the changes in the revised manuscript.
Point 12:
"Most of the informal caregivers had at least one significant training need". How much is "most"? Moreover, how many had two or more needs? Please add all this information.
Response 12:
Thank you for this important suggestion.
In our survey, 86.7% (N = 170) of caregivers indicated that their training needs were not being fully met and reported at least one need for targeted training activity, and 62.7% (N = 123) of caregivers identified two or more training needs.
We added all this information in the abstract. Please see the changes in the revised manuscript.
Point 13:
"Factors associated with technical skills training needs were primarily care recipients’ diseases/conditions and caring-related variables." A very vague sentence. Please give examples.
Response 13:
Thank you for this important comment and suggestion.
In response to your suggestion, we have revised the expression and added some specific examples. Please see the changes in the revised manuscript.
INTRODUCTION
Point 14:
"Influenced by intergenerational compensation and traditional family culture, informal care mainly provided by family members has long been the primary way to meet the care needs of the elderly in China". Please elaborate better on this issue as it provides a motivation for your study.
Response 14:
Thank you for this valuable suggestion.
We have rephrased this paragraph to highlight the important role of informal caregiving in the long-term care system. Please see the changes in the revised manuscript.
Point 15:
"With the development of China's rapidly expanding elderly population" - what does this collection of words mean? Please rephrase.
Response 15:
Thank you for pointing this out to us.
The point we're trying to express is that “As the aging population continues to grow”.
We have fixed this sentence, please see the change in the revised manuscript.
Point 16:
Line 46 - please replace the word "preparation" with "training".
Response 16:
Thank you for this helpful suggestion.
According to your suggestion, we have replaced the word “preparation” with “training”. Please see the change in the revised manuscript.
Point 17:
"When these high-intensity caregiving tasks are overlain with significant caregiving skill gaps, caregivers can experience substantial stress and markedly decrease the physical and mental health". What is the mechanism? Why would people experience stress and burden? What does caregiver stress model maintain about that? Please discuss all this.
Response 17:
Thank you for this insightful comment and suggestions.
Numerous studies have demonstrated and explained the correlation between informal caregiving and health status. Among them, the stress process model was widely used. Specifically, the complex caregiving needs of care recipient are often considered as a primarily stressor, leading to subjective caregiving burden for caregivers, either directly or through influencing their caregiving behaviors (time spent caring). When informal caregivers are burdened or overloaded for a long period of time, it can undoubtedly have a significant negative impact on their physical and mental health. Of course, all steps in this process are influenced by pre-existing personal characteristics (e.g. caregiving experience, skills) and concurrent buffering factors (e.g. social support).
We have discussed all of these issues in the introduction section. Please see the changes in the revised manuscript.
Point 18:
"In China, however, informal caregivers contribute the bulk of long-term care services, but with little access to supportive services for themselves." But what are the existing nationwide intiatives? Please outline.
Response 18:
Thank you for this valuable suggestion.
In China, nationwide initiatives preferred to support informal caregivers indirectly by providing financial and service assistance to the care recipients. Examples include the provision of living allowances, home care services, and long-term care insurance for older adults. Supportive policies directly targeting informal caregivers remain limited, such as individual income tax deductions for children supporting their parents, and nursing leave for one-child families. There are no national policies in place to counseling and training for informal caregivers.
We have added the content above. Please see the change in the revised manuscript.
Point 19:
The entire section provides absolutely no relation to the previous research in the field. Why would the scientific community need this study? What does it discover that the others did not succeed in? Please elaborate on these issues using the articles I cited above and the other ones.
Response 19:
Thank you for your insightful comments and suggestions.
Due to the high-intensity caregiving tasks overlaid with significant caregiving skill gaps, informal caregivers are consequently caregivers are consequently exposed to high levels of stress and, ultimately, deteriorating physical and mental health. And quality skills training holds great promise in improving the situation of informal caregivers. Therefore, it is necessary to understand the characteristics of training needs of informal caregivers so as to contribute to the development of better training interventions. Currently, an increasing number of studies have begun to focus on the training needs of informal caregivers. However, on the one hand, existing studies have not focused on the correlation between caregiving support and training needs of informal caregivers, and on the other hand, fewer studies have been conducted on informal caregiving training needs and their influencing factors based on Shanghai, China. This paper focuses on the correlation between different types of social support and training needs of informal caregivers in Shanghai, to some extent bridging the gap of existing research.
We have re-described it in detail in the introduction section. Please see the change in the revised manuscript.
Point 20:
Who is going to benefit from the study findings and in which ways? Please dedicate a paragraph for this.
Response 20:
Thank you for this valuable suggestion.
Our findings are expected to provide an important reference value for policy makers to design effective caregiving training strategies. And appropriate training interventions holds great promise in reducing caregiving burden and improving health status of informal caregivers. In addition, it is also expected to improve the health status of care recipients, increase their ability to remain safely in their own communities, and thus alleviate the burden on health care systems.
We have added these in the introduction section. Please see the change in the revised manuscript.
MATERIALS AND METHODS
Point 21:
"The fourth iteration of the “Elderly Helping Program” was implemented in 2021," When were the three first iterations held? Please add.
Response 21:
Thank you for this helpful suggestion.
Shanghai implemented the first iteration of the “Elderly Helping Program” in 2018. Since then, the program has been conducted annually, with four iterations by 2021. With the development of the program, its coverage has been gradually expanded, from the initial 11 sub-districts to 45 sub-districts in the fourth iteration.
In response to your suggestion, we have added a description of the evolution of the “Elderly Helping Program”. Please see the change in the revised manuscript.
Point 22:
"...a set of relevant conditions," What were they? Please add.
Response 22:
Thank you for pointing this out to us.
The conditions are informal caregivers who engaged in the “Elderly Helping Program” during 2021, and were caring for an older adult with any ADL/IADL impairment.
According to your suggestion, we have added this content. Please see the change in the revised manuscript.
Point 23:
Line 99 - Why the 16-year bound was chosen as the lowest limit? Please justify in the text.
Response 23:
Thank you for this valuable question.
The lowest age limit for eligible informal caregivers was chosen to be 16-year bound, mainly because in China's national context, a person aged 16-year-old and above is considered to have full capacity for civil behavior and has completed nine years of compulsory education. So, we believe that he/she is able to take on the role of an informal caregiver and has the cognitive ability to understand their training needs. In addition, given that informal caregivers are mostly family members, and that there are likely to be cases of younger grandchildren caring for their grandparents, we eventually chose a broader age boundary.
Point 24:
"the primary informal caregiver". How was this identified? Please add.
Response 24:
Thank you for this important suggestion.
The primary informal caregivers should be those who “have spent at least 1 hour per day performing care tasks or provided care consistently for at least 1 month (a minimum of 1 time per week during that month)”. These were mentioned in detail in the respondent inclusion criteria.
This was mainly to avoid some family members living with the disabled older adults (who may actually provide little or no care) being defaulted to informal caregivers, which may result in statistical bias (those who have only transient experiences of informal caregiving may be unaware of what they need in terms of training).
We realized that the presentation of the participants screening criteria was not clear. Therefore, we further fully explained the criteria by numbering them one by one. Please see the changes in the revised manuscript.
Point 25:
"with any physical activity impairment" How did you define it? Please add.
Response 25:
Thank you for this helpful suggestion.
A care recipient is considered to have a physical activity impairment when he/she is unable to perform any of the ADLs or IADLs independently and requires certain assistance.
Given the lack of clarify of the phrase “with any physical activity impairment”, we have revised it to “with any ADL/IADL impairment”. We hope that it will more accurately reflect what we intended to express.
Please see the change in the revised manuscript.
Point 26:
How was the "consistently" (line 101) defined? Please add.
Response 26:
Thank you for this helpful suggestion.
The “consistently” was defined as providing informal care at least once a week for a period of one month.
We have added the explanation of the term “consistently”. Please see the change in the revised manuscript.
Point 27:
"A total of 268 eligible informal caregivers participated in the survey." Out of how many approached/filtered out? Please provide a full flow of the participants, also using a figure.
Response 27:
Thank you for this important suggestion.
By setting a series of respondent inclusion criteria, 37 caregivers were filtered out, and a total of 268 eligible informal caregivers voluntarily participated in the survey. Among them, 196 valid sample were obtained for this research, with a sample validity of 73.13%.
According to your valuable suggestion, we have provided a full flow of the participants in the revision. Please see the change in the revised manuscript.
Point 28:
"Face-to-face questionnaire-based surveys were administered during December 3, 2021 to January 23, 2022.". In which settings? How was the entire process of interviewing organized? Why it was decided to stop the data collection on the second date mentioned? Why FTF interviews were chosen? Please add all this information. Please also replace the words "during... to" with "between.... and ...."
Response 28:
Thank you for these valuable suggestions.
The survey was based on a government-sponsored evaluation of the “Elderly Helping Program”, and informal caregivers were invited to be interviewed at their homes or training centers with the assistance of the training organizations. Following the timeframe of the program evaluation, the whole survey was implemented between 3 December 2021 and 23 January 2022. Meanwhile, face-to-face paper version questionnaires were used to improve the validity and reliability of the survey, as well as to better identify the real unmet training needs of informal caregivers.
We feel that your suggestions have helped us to better describe the entire process of the survey. And based on your suggestions, we have added these above. Please see the changes in the revised manuscript.
Point 29:
Were the education and the income variables used in a continuous form or were dichotomized? Please better elaborate here. If the former is right, please explain why you treated the three-category variable as such.
Response 29:
Thank you for this valuable suggestion.
Considering the possible nonlinear relationship between the multi-category variables and the dependent variables, the real relation between the independent variables and the dependent variables may not be better reflected if a continuous form is used. Therefore, both education and income variables were converted into dummy variables in the correlation analysis.
Details about the operationalization of the education and income variables have been added in the revision. Please see the change in the revised manuscript.
Point 30:
How were the other variables defined for the purposes of the regression analysis? Please add the detailed information.
Response 30:
Thank you for this helpful suggestion.
The detailed information of the other variables has been added in the methods section. Please see the changes in the revised manuscript.
Point 31:
I do not understand what the "hierarchical relationship" (line 130) means. Please rephrase.
Response 31:
Thak you for this valuable question.
When measuring physical impairments in older adults, some of the literature tends to combine ADL with IADL. However, we argue that ADL and IADL reflect different dimensions of mobility, respectively, and to some extent have a progressive relationship. Therefore, it is necessary to include them as two separate variables in the correlation analysis. Specifically, on the one hand, relative to ADL, IADL include more complex activities related to independent living in the community, and thus, caregiver training needs may differ between the ADL and IADL levels of the care recipients. On the other hand, ADL impairments usually imply more severe disability and greater intensity of caregiving burden than IADL, and caregiver training needs may also vary across the intensity of caregiving burden.
In order to avoid unnecessary misunderstanding, we have deleted the phrase “hierarchical relationship”. Please see the change in the revised manuscript.
RESULTS
Point 32:
Table 1 - it is unclear what the statistics for the need variables mean. Please show clear and comprehensible measures.
Response 32:
Thank you for pointing this out to us.
Categorical variables should report numbers and percentages in the descriptive statistics. However, we presented the mean and standard deviation in the original manuscript.
We have deleted the statistics of the need variables in Table 1 , and instead expressed the statistical parameters of the need variables through figure 1. Please see the changes in the revised manuscript.
Point 33:
Please divide Figure 1 into two separate ones as there is currently a lot of information inside. Please also arrange the figures in some order (either descending or ascending).
Response 33:
Thank you for this helpful suggestion.
We have divided Figure 1 into two separate ones, and also arranged the figures in descending order. Please see the changes in the revised manuscript.
Point 34:
The reason for the separation between the tables is unclear. If you still decide to do this after reexamination of the models, as was asked above, please justify this separation.
Response 34:
Thank you for this helpful suggestion.
The main reason we separated the tables was that we wanted to separately elaborate on the factors influencing technical training needs versus intangible training needs. However, this seemed to increase the confusion of the article, so, following your suggestion, we have combined the two tables. Please see the change in the revised manuscript.
LIMITATIONS
Point 35:
There are two meaningful limitations that were not mentioned. First, a relatively small sample. Second, the potential relevance of the findings for the studied period only. Note that the data was collected during the pandemic.
Response 35:
Thank you for your insightful comment.
We have re-written the limitations section according to your suggestions, adding limitations regarding the sample size and studied period. Please see the changes in the revised manuscript.
Comments on the Quality of English Language
Point 36:
A quite poor one. There are too many awkwardly or incorrectly sounding parts of sentences (e.g. "…does not take informal caregivers' demand substantially into consideration." or "...well-suited intervention programs."), irrelevant or misused words and expressions. A thorough proofreading is essential.
Response 36:
Thank you for pointing this out to us.
We thoroughly proofread the English grammar and phrasing of the article, and a professional language editor assisted us with the modified manuscript. We hope that the grammar and the phrasing are now acceptable.

Reviewer 4 Report
Comments and Suggestions for Authors
This manuscript titled "Perceived training needs of informal caregivers of the elderly: A Cross-sectional Study. Overall the manuscript needs to be clarified.
Introduction: what was the hypothesis of the study?
Materials and Methods:
Did you calculate the sample size? Please also provide the sampling. technique.
What is the definition of an informal caregiver?
Line 188-189: You stated that "To avoid including confounding factors or missing important factors, a p value < 0.200 was adopted in the univariate analyses (Table A1)." Why did you set a p-value as < 0.20? why it is not p<.05
Discussion: Line 280-281, you stated "The results reveal that informal caregiving in Shanghai is labor-intensive, with an average of approximately 28 hours of caring activities provided per week." do you have evidence support or reference to support the average of caring activities per week?
Comments on the Quality of English Language
English is fine.
Author Response
Response to Reviewer 4 Comments and Suggestions
This manuscript titled "Perceived training needs of informal caregivers of the elderly: A Cross-sectional Study. Overall the manuscript needs to be clarified.
Thank you for providing us with the valuable opportunity to modify our manuscript and resubmit it. Your suggestions and comments are very helpful. We have greatly improved the manuscript according to your important and enlightening suggestions. We hope that the manuscript is now ready for publication.
Introduction
Point 1:
what was the hypothesis of the study?
Response 1:
Thank you for this important question.
This paper aims to explore the potential training needs of informal caregivers, and to identify the factors associated with these perceived training needs. In addition to general demographic factors, we were particularly interested in understanding whether informal caregivers' caregiving burdens (e.g. relationship to the care recipient, caregiving time) and caregiving support resources (e.g. formal care support, informal care support) were associated with different perceived training needs.
Therefore, the hypotheses of our study were that:
- physical/psychological caregiving burden is significantly associated with different types of perceived training needs of informal caregivers;
- formal/informal care support is significantly associated with t different types of perceived training needs of informal caregivers.
It should be noted that our research hypotheses are relatively broad as our study is exploratory and involves only the exploration of factors related to training needs.
Thank you very much for your enlightening comments, and following your suggestions, we have added a note about the study hypothesis in the method section. Please see the change in the revied manuscript.
Materials and Methods
Point 2:
Did you calculate the sample size? Please also provide the sampling technique.
Response 2:
Thank you for the valuable suggestion.
First, sample size calculation was conducted to ensure enough data was collected. We referred to Kendall sample estimation method, which proposes the sample size for multivariate analysis should be 5-10 times of the predictor variables. In our study, a total of 20 predictor variables were considered, and we estimated that 10%-20% of the questionnaires were invalid. Thus, the reasonable sample size is 112 <= N <= 250, which indicated our sample was adequate.
Second, for the sampling technique, a multi-stage cluster random sampling method was used to randomly select 10 sub-districts out of the 45 sub-districts where the program was implemented in 2021 as the investigation sites. All informal caregivers of older adults who engaged in the training program during 2021 were invited to participant in the survey.
In summary, we have tried our best to collect a sample that is generally representative of informal caregivers of the disabled older adult in Shanghai.
Thank you for improving the clarity of our manuscript. It is very important for the readers to be aware of the sample size and sampling technique, especially when a limited number of participants are included. The sampling technique and the calculation of reasonable sample size have been added in the method section. Moreover, we discussed this sample size problem in the limitation section, suggesting that there is still room for further investigation with larger sample size.
Point 3:
What is the definition of an informal caregiver?
Response 3:
Actually, the precise definition of the term “informal caregiver” is inconsistent across studies, with the core controversies centering on “relationship”, “payment”, and “professionalism”. Synthesis of definitions from existing studies, we consider an “informal caregiver” to be: (1) a person whose relationship with care recipient is predominantly family like spouses, children or children-in-law, but not except neighbors, friends, and volunteers; (2) a person who does not receive any payment from the service organizations for the care provided (except “Timing banking program”); (3) a person who may has received some level of informal training (usually through seminars, workshops, etc.), but exclude those who are formally trained, registered and have professional credentials.
References:
- edSchulz R, Tompkins CA. Chapter: 7. Informal Caregivers in the United States: Prevalence, Caregiver Characteristics, and Ability to Provide Care. In: The role of human factors in home health care: workshop summary. Washington DC, 2010.
- Schultz R, Martire LM. Family caregiving of persons with dementia: Prevalence, health effects, and support strategies. Am J Geriatr Psychiatry. 2004;12(2):240-249.
- Wenwei Liu, Tongzhou Lyu et al. Willingness-to -pay and willingness-to-accept of informal caregivers of dependent elderly people in Shanghai, China. BMC Health Services Research, 2020,20:618.
- Adashek JJ, Subbiah IM. Caring for the caregiver: a systematic review characterising the experience of caregivers of older adults with advanced cancers. ESMO Open. 2020;5(5):e000862. doi:10.1136/esmoopen-2020-000862
Point 4:
Line 188-189: You stated that "To avoid including confounding factors or missing important factors, a p value < 0.200 was adopted in the univariate analyses (Table A1)." Why did you set a p-value as < 0.20? why it is not p<.05
Response 4:
Thank you for this insightful question.
As you note, the selection parameter (that is, the significance level to decide whether an effect should be retained in the model) must be carefully chosen when screening candidate predictor variables through univariate analysis. On this issue, most researchers tend to believe that too strict p-value settings are likely to lead to the omission of important adjustment variables that need to be controlled in the multivariate model. Smaller values such as 0.05 or 0.01 are only recommended for very large sample sizes (EPV of 100 or above). In the vast majority of applications, especially when the regression model does not have many predictor variables and the sample size is not that large, a value of 0.20 or 0.157 (corresponding to selection based on AIC) or even 0.5 (resulting in very mild selection) will be a better choice (Heinze G, Dunkler D, 2017).
Therefore, we established a more moderate parameter selection (p < 0.20) in the univariate analysis. Interestingly, it is found that some predictor variables did not pass the significant level test of p < 0.05 in the univariate analysis, but showed a stronger correlation with the core explanatory variables in the multivariate analysis (p < 0.05).
We further fully explained the significance level setting for the selection of predictor variables through univariate analysis in the method section. Please see the changes in the revised manuscript.
Reference:
Heinze G, Dunkler D. Five myths about variable selection. Transplant International. 2017;30(1):6-10. doi:10.1111/tri.12895
Discussion
Point 5:
Line 280-281, you stated "The results reveal that informal caregiving in Shanghai is labor-intensive, with an average of approximately 28 hours of caring activities provided per week." do you have evidence support or reference to support the average of caring activities per week?
Response 5:
Thank you for this important question.
There is relatively limited research on informal care in China, and few studies have counted the average weekly caregiving hours for informal caregivers. A global survey by Merck supports our findings. The Embracing Carers initiative, launched by Merck in partnership with the International Alliance of Caregiver Organizations (IACO), published the Carer Wellbeing Index in 2021. The survey included more than 9,000 carers from 12 countries in Asia, Europe and the Americas, including more than 750 from China. These caregivers are providing informal care for loved ones with long-term chronic illnesses, physical disabilities, or cognitive/mental illnesses. According to statistics, the average weekly caregiving hours of caregivers in China is 25.3 hours, second only to the 28 hours in the United States, and close to the average weekly caregiving hours in the United Kingdom (25.9 hours) and Brazil (25.1 hours), and much higher than that of countries such as France (18.1 hours) and Germany (17.8 hours). Our study sample shows that the average weekly caregiving hours of informal caregivers in Shanghai is 28.02 hours, which is slightly higher than the results of the study by Mercer.
In terms of average daily caregiving hours, according to established standards, informal caregivers caring for less than 4 hours per day are considered as low-intensity care, 4 to 8 hours are medium-intensity, and more than 8 hours are high-intensity care (Bremer et al., 2015). In contrast, in our survey sample, 28.3% (N = 75) cared for less than 4 hours per day, 53% (N = 104) cared for 4 to 8 hours, and 9.7% (N = 17) cared for more than 8 hours. This indicates that the majority of informal caregivers have a high intensity of caregiving.
Supportive evidence on the average of caregiving has been added to the article. Please see the change in the revised manuscript.
References
- Embracing caregivers, Merck in action - focusing on the informal caregiver in COVID-19 epidemic. 2021. Accessed March 29, 2023. https://www.sohu.com/a/460771359_100207642
- Bremer P, Cabrera E, Leino – Kilpi H, et al. informal dementia care: consequences for caregivers’ health and health care use in 8 European countries. Health Policy. 2015;119(11):1459-1471.

Round 2
Reviewer 3 Report
Comments and Suggestions for Authors
I would like to thank the authors for a very thorough approach taken to revise the manuscript.
Most of my comments in the previous round were addressed in a satisfactory way. However, some of them were still not, and there is also some greater work necessary in order to consider this manuscript for publication again.
I provide my comments in accordance with the flow of the article:
ABSTRACT:
- Here you provide the sample size of 268 respondents. However, as was revealed in the Methods chapter, this number reduced to 196. Please replace and rewrite the results here accordingly.
- Lines 30-31 - The sentence included in them sounds unclear. Please rewrite.
- Line 37 - Do you really think one would type "health and functional-oriented skills" to seek for the literature? Please rephrase so this keyword would contain two words only.
INTRODUCTION
- Lines 44-45 - The sentence that begins with "And" sounds incomplete. Please modify.
- In addition, I do not know who was your or your editor's English teacher but what he/she did not teach you is that English sentence does not start with the word "And". Please rephrase all related sentences that start this way (for instance, in lines 67, 138, and more).
- Line 46 - Please replace the word "begin" with "begun" to be consistent with the tense employed in the previous line.
- Lines 51-52 - Please support the claim outlined in these lines by the literature.
- Line 55 - Please delete the word "are".
- Line 58 - What do you mean by health? Health status? Self-rated health? Physical health? Mental health? Please be more precise here.
- Line 58 - Who proposed/developed this model? Please cite.
- Line 63 - In academic English, sentences do not start with "Of course". Please rephrase.
- Lines 99-103 - A very long sentence. Please split it.
- Lines 104-106 - Please cite those studies.
- Line 117 - There is no need to start the word "nationwide" with capital letter. Please amend.
- Line 136 - Unclear what the "caregiving supports" mean. Please rephrase.
- Line 140 - You do not have to use the word "also" when you begin the sentence with "In addition". Please remove.
MATERIALS AND METHODS
- Line 154 - Please remove the word "be" that stands before listing the first exclusion criterion.
- Lines 164-166 - Please rephrase this sentence as it sounds unclear.
- Lines 178-179 - Please support this claim by the related literature.
- Lines 233-234 - Does the literature support this decision of yours? Please cite.
- Lines 240-241 - What was the reference category? Please add.
- Lines 264-265 - What was the reference category for each of the variables? Please add.
RESULTS
- The numbers provided inside the bars in both Figure 1 and 2 are not clearly seen. Please mark them with a bright color.
- Table 2 must be reorganized. I counted 15 variables in total that were associated with having any need at least once. Please reorganize the table so that it would be clearly seen which independent variable is associated with which dependent variable (and with which it does not). In addition, please add the note for the meaning of the asterisks.
DISCUSSION
- This chapter should contain more relation to the previous literature. What does each finding suggest? Do they support/contradict the existing research? Please better elaborate.
- The relation to each association found must be better presented. I would suggest going from the most dominant variable (ADL seems like one) to the least dominating ones. Please elaborate.
Comments on the Quality of English LanguageSome English proofreading is necessary
Author Response
Response to Reviewer 3 Comments
I would like to thank the authors for a very thorough approach taken to revise the manuscript.
Most of my comments in the previous round were addressed in a satisfactory way. However, some of them were still not, and there is also some greater work necessary in order to consider this manuscript for publication again.
I provide my comments in accordance with the flow of the article:
Thank you for providing us with the valuable opportunity to modify our manuscript and resubmit it. Your suggestions and comments are very helpful. We have greatly improved the manuscript according to your important and enlightening suggestions. We hope that the manuscript is now ready for publication.
ABSTRACT:
Point 1:
- Here you provide the sample size of 268 respondents. However, as was revealed in the Methods chapter, this number reduced to 196. Please replace and rewrite the results here accordingly.
Response 1:
Thank you for pointing it out to us.
We have rewritten the sample size to be consistent with the results in the Methods chapter. Please see the change in the revised manuscript (line 23, page 1).
Point 2:
- Lines 30-31 - The sentence included in them sounds unclear. Please rewrite.
Response 2:
Thank you for this important suggestion.
We have modified this sentence. Please see the change in the revised manuscript (line 28-33, page 1).
Point 3:
- Line 37 - Do you really think one would type "health and functional-oriented skills" to seek for the literature? Please rephrase so this keyword would contain two words only.
Response 3:
Thank you for this insightful comment and suggestion.
After consideration, we have replaced this keyword “health and functional-oriented skills” with “health oriented”, which is the key concept we want to emphasize. We have also conducted a web of science search based on this new keyword and have found a large amount of relevant literature.
Please see the change in the revised manuscript (line 38, page 1).
INTRODUCTION
Point 4:
- Lines 44-45 - The sentence that begins with "And" sounds incomplete. Please modify.
- In addition, I do not know who was your or your editor's English teacher but what he/she did not teach you is that English sentence does not start with the word "And". Please rephrase all related sentences that start this way (for instance, in lines 67, 138, and more).
Response 4:
Thank you for pointing this out to us.
We have modified this mistake. Please see the change in the revised manuscript (line 42, page 1). We have also checked the full text and have rephrased all related sentences that start with the word “And”.
In order to avoid similar English grammar mistakes, we have re-invited a professional language editor to touch up the whole text, and we hope that the grammar and the phrasing are now acceptable.
Point 5:
- Line 46 - Please replace the word "begin" with "begun" to be consistent with the tense employed in the previous line.
Response 5:
Thank you for your helpful suggestion.
We have replaced the word “begin” with “begun” to be consistent with the tense employed in the previous line. Please see the change in the revised manuscript (line 44, page 1).
Point 6:
- Lines 51-52 - Please support the claim outlined in these lines by the literature.
Response 6:
Thank you for this important suggestion.
Following your suggestion, we have provided a supporting literature basis for this claim.
Zarzycki M. et al.’s study provides important support for our claim. Zarzycki M. et al. (2023) conducted a systematic review on the motivations of informal caregivers to provide care and argued that cultural and societal factors strongly underpinned motivations and willingness for informal caregiving.
Point 7:
- Line 55 - Please delete the word "are".
Response 7:
Thank you for pointing it out to us. We have deleted this word to make the sentence more reasonable. Please see the change in the revised manuscript.
Point 8:
- Line 58 - What do you mean by health? Health status? Self-rated health? Physical health? Mental health? Please be more precise here.
Response 8:
Thank you for pointing it out to us.
Numerous studies have demonstrated the correlation between informal caregiving and carers’ health status, indicating that excessive caregiving can undoubtedly have a significant negative impact on carers’ physical and mental health.
We have clarified exactly what we mean by “health”. Please see the change in the revised manuscript (line 52, page 2).
Point 9:
- Line 58 - Who proposed/developed this model? Please cite.
Response 9:
Thank you for this helpful suggestion.
In the 1980s, Pearlin et al. proposed the framework of the Stress Process Model, based on theories of stress coping, to identify the mechanisms linking stressors and outcomes.
We have indicated who proposed the Stress Process Model. Please see the change in the revised manuscript (line 53, page 2).
Point 10:
- Line 63 - In academic English, sentences do not start with "Of course". Please rephrase.
Response 10:
Thank you for pointing this out to us.
We have modified the phrasing of this sentence. Please see the change in the revised manuscript (line 58, page 2).
Point 11:
- Lines 99-103 - A very long sentence. Please split it.
Response 11:
Thank you for pointing this out to us.
We have split this long sentence into two relatively short sentences. Please see the change in the revised manuscript (line 88-92, page 2).
Point 12:
- Lines 104-106 - Please cite those studies.
Response 12:
Thank you for your helpful suggestion.
We have added relevant literature citations. Please see the change in the revised manuscript (line 96, page 2).
Point 13:
- Line 117 - There is no need to start the word "nationwide" with capital letter. Please amend.
Response 13:
Thank you for pointing this out to us.
We have amended this error. Please see the change in the revised manuscript (line 110, page 3).
Point 14:
- Line 136 - Unclear what the "caregiving supports" mean. Please rephrase.
Response 14:
Thank you for this important suggestion.
“Caregiving support” refers to other complementary care services received by informal caregivers during the provision of care. We consider two sources of caregiving support. The first source is the use of formal care support, i.e. professional caregiver support from the government or the market. Other informal care support is the second source of support. This measure indicates whether the care recipient had support from other nonprofessional caregivers than the one interviewed, including family member, friend, neighbor, acquaintance, and volunteer.
We have rephrased “caregiving supports” to “complementary caregiving supports (e.g., formal care and other informal care)”. Please see the change in the revised manuscript (line 128-129, page 3).
Point 15:
- Line 140 - You do not have to use the word "also" when you begin the sentence with "In addition". Please remove.
Response 15:
Thank you for pointing this out to us.
We have amended this error. Please see the change in the revised manuscript (line 131-134, page 3).
MATERIALS AND METHODS
Point 16:
- Line 154 - Please remove the word "be" that stands before listing the first exclusion criterion.
Response 16:
Thank you for pointing this out to us.
We have removed the word “be” that stands before listing the first exclusion criterion. Please see the change in the revised manuscript (line 146, page 3).
Point 17:
- Lines 164-166 - Please rephrase this sentence as it sounds unclear.
Response 16:
Thank you for pointing this out to us.
We have modified the formulation of this sentence. Please see the change in the revised manuscript (line 156-157, page 4).
Point 18:
- Lines 178-179 - Please support this claim by the related literature.
Response 18:
Thank you for this helpful suggestion.
We have provided a literature citation for this claim. Please see the change in the revised manuscript (line 169-171, page 4).
Point 19:
- Lines 233-234 - Does the literature support this decision of yours? Please cite.
Response 19:
Thank you for this insightful question.
Through reviewing the literature, we found that most of the existing measures of caregiving burden are based on well-established burden scales, such as the CBI (Caregiver Burden Inventory), ZBI (Zarit Burden Interview), CRA (Caregiver Reaction Assessment), CSI (Caregiver Strain Index), BSFC (the Burden Scale for Family Caregivers), etc. There have been no studies using caregiving hours as a proxy variable for physiologic caregiving burden.
A further review of the literature on the caregiving burden found that, for the most part, variables such as caregiving time, and relationship to the care recipient were more likely to be considered as objective caregiving stressors (Zhou & Yan, 2021). The bearing, perception and evaluation of different stressors form the caregiver's subjective caregiving burden (Pearlin, 1990). Based on these findings, we used the terms “caregiving time” and “relationship to care recipient” to refer to “physiological caregiving stressor” and “emotional caregiving stressor”, respectively. Evidence on the correlation between caregiving time and physiological burden and caregivers’ physical health (Zhong, 2010) provides a strong theoretical basis for using “caregiving time” to refer to “physiological caregiving stressor” in this paper. Evidence on the correlation between identity structure and psychological burden and caregivers’ mental health (Bevans & Sternberg, 2012) also provides a strong rationale for the use of “relationship to care recipient” to refer to “emotional caregiving stressors”.
Finally, a large body of research has demonstrated the mechanisms between caregiving stressors, caregiving burden, and behavioral outcomes. The propensity for training needs is one of the possible behavioral outcomes for caregivers in the face of caregiving stressors as well as caregiving burden. Therefore, the relationship between caregiving stressors and training needs should be fully explored.
In response to your insightful suggestions, we have made further changes and added literature citations in the revisions. Please see the revised manuscript for more details (line 223-232, page 5).
Point 20:
- Lines 240-241 - What was the reference category? Please add.
Response 20:
Thank you for pointing this out to us.
We have added a description of the reference category. Please see the change in the revised manuscript (line 234-235, page 5).
Point 21:
- Lines 264-265 - What was the reference category for each of the variables? Please add.
Response 21:
Thank you for pointing this out to us.
We have added a description of the reference category for each of the variables. Please see the change in table 2 of the revised manuscript.
RESULTS
Point 22:
- The numbers provided inside the bars in both Figure 1 and 2 are not clearly seen. Please mark them with a bright color.
Response 22:
Thank you for this helpful suggestion.
Following your suggestion, we have marked the numbers with a bright color. Please see the changes in the revised manuscript.
Point 23:
- Table 2 must be reorganized. I counted 15 variables in total that were associated with having any need at least once. Please reorganize the table so that it would be clearly seen which independent variable is associated with which dependent variable (and with which it does not). In addition, please add the note for the meaning of the asterisks.
Response 23:
Thank you for this important suggestion.
We have reorganized Table 2 and added the note for the meaning of the asterisks. We hope that the new table format can clearly show the relationship between the core independent variables and dependent variables.
Please see the changed in the revised manuscript (page 10-11).
DISCUSSION
Point 24:
- This chapter should contain more relation to the previous literature. What does each finding suggest? Do they support/contradict the existing research? Please better elaborate.
Response 24:
Thank you for your insightful suggestions.
Based on your suggestions, we have re-incorporated additional references to support our findings. In addition, we have further elaborated our findings, including what they imply, what valuable inputs they can provide to current informal care system, and so on. We hope that this revision will further deepen the value of our study.
Please see the discussion section of the revised manuscript for more details.
Point 25:
- The relation to each association found must be better presented. I would suggest going from the most dominant variable (ADL seems like one) to the least dominating ones. Please elaborate.
Response 25:
Thank you for your insightful suggestions.
We have reorganized the logic of this section and, based on your suggestions, have better addressed each of these relations one by one, starting with the most dominant variables (health status of the care recipients).
According to their importance and attributes, the relations have been organized into four modules, namely, “health status of care recipient”, “caregiving support resources”, “Caregiving Stressors”, and “personal attributes of Caregiver”.
Please see the discussion section of the revised manuscript for more details.
Comments on the Quality of English Language
Point 26:
Some English proofreading is necessary
Response 26:
Thank you for pointing this out to us.
We thoroughly proofread the English grammar and phrasing of the article. We hope that the grammar and the phrasing are now acceptable.
Reviewer 4 Report
Comments and Suggestions for Authors
The revised manuscript can be accepted.
Comments on the Quality of English LanguageMinor editing of English language required.
Author Response
Thank you for taking the time to review our manuscript. Your approval of our manuscript gives us great confidence. We have made a second round of revisions based on comments from other reviewers, please see the revised manuscript for more details. Thank you again for your kind help!